# Lightweight Transformer for EEG Classification via Balanced Signed Graph Algorithm Unrolling

**Junyi Yao**
Peking University
2401112160@stu.pku.edu.cn

**Parham Eftekhar**
York University
eftekhar@yorku.ca

**Gene Cheung**
York University
genec@yorku.ca

**Xujin Chris Liu**
New York University
xl3942@nyu.edu

**Yao Wang**
New York University
yw523@nyu.edu

**Wei Hu**[*]
Peking University
forhuwei@pku.edu.cn

## Abstract

Samples of brain signals collected by EEG sensors have inherent anti-correlations that are well modeled by negative edges in a finite graph. To differentiate epilepsy patients from healthy subjects using collected EEG signals, we build lightweight and interpretable transformer-like neural nets by unrolling a spectral denoising algorithm for signals on a balanced signed graph—graph with no cycles of odd number of negative edges. A balanced signed graph has well-defined frequencies that map to a corresponding positive graph via similarity transform of the graph Laplacian matrices. We implement an ideal low-pass filter efficiently on the mapped positive graph via Lanczos approximation, where the optimal cutoff frequency is learned from data. Given that two balanced signed graph denoisers learn posterior probabilities of two different signal classes during training, we evaluate their reconstruction errors for binary classification of EEG signals. Experiments show that our method achieves classification performance comparable to representative deep learning schemes, while employing dramatically fewer parameters.

## 1 Introduction

We study the classification of EEG signals in patients with epilepsy versus healthy control subjects. Compared to classical model-based methods, such as k-Nearest Neighbors with dynamic time warping features (Tasci et al., 2023) and feature extraction from time–frequency maps (Shen et al., 2024), *deep learning* (DL) models, such as CNN-based Shen et al. (2024), Bhandage et al. (2024) and Dicsli et al. (2025) and more recent transformer-based Lih et al. (2023), have achieved state-of-the-art (SOTA) results (*e.g.*, up to the 90% range). However, the transformer model consumes an enormous number of parameters and functions as an uninterpretable black box. Thus, *parameter reduction* and *interpretation* of learning models is crucial towards practical implementation on resource-constrained EEG devices.

An alternative paradigm for data learning is *algorithm unrolling* (Monga et al., 2021): first design an iterative optimization algorithm minimizing a mathematically-defined objective, then "unroll" each iteration into a neural layer, and stack them back-to-back to compose a feed-forward network for data-driven parameter learning. Notably, Yu et al. (2023) recently unrolls an algorithm minimizing a *sparse rate-reduction* (SRR) objective into a transformer-like neural net—called "white-box transformer"—that achieves comparable performance as SOTA in image classification, while remaining 100% mathematical interpretable[1].

---

[*]Corresponding author.

[1]Common in algorithm unrolling (Monga et al., 2021), "interpretability" here means that each neural layer corresponds to an iteration of an optimization algorithm minimizing a mathematically-defined objective.

Inspired by Yu et al. (2023), for the EEG signal classification problem we also build transformers via algorithm unrolling, but from a unique *graph signal processing* (GSP) perspective (Ortega et al., 2018; Cheung et al., 2018). GSP studies mathematical tools such as transforms, wavelets, and filters for discrete signals residing on irregular data kernels described by graphs. Recently, Thuc et al. (2024) shows that a graph learning module with edge weight normalization plays the role of self-attention (Bahdanau et al., 2014), and thus unrolling a graph algorithm with graph learning modules inserted yields a transformer-like neural net. However, Thuc et al. (2024) focuses solely on *positive* graphs that model simple pairwise *positive* correlations among neighboring pixels in a static image.

For EEG signals, collected samples often exhibit pairwise anti-correlations, which are effectively modeled by *negative* edges. Though frequencies for general signed graphs (with both positive and negative edges) are not well understood, Dinesh et al. (2025) shows that in the special case of *balanced signed graphs*—with no cycles of odd number of negative edges—frequencies can be rigorously defined: the Laplacian matrix $\mathcal{L}^B$ of a balanced signed graph $\mathcal{G}^B$ is a similarity transform of the Laplacian $\mathcal{L}^+$ of a corresponding positive graph $\mathcal{G}^+$ (hence they share the same eigenvalues), and the spectra of positive graphs are well understood and utilized in GSP (Ortega et al., 2018). Thus, widely studied filters for positive graphs (Onuki et al., 2016; Shuman, 2020) can be readily reused for signals on balanced signed graphs (Yokota et al., 2025).

We leverage this fact to build EEG signal denoisers $\mathbf{\Psi}(\cdot)$ as a *pretext task*[2] for later binary classification. Specifically, we first learn a balanced signed graph $\mathcal{G}^B$ from EEG data; graph balance is ensured during signed edge weight assignment via a novel interpretation of the *Cartwright-Harary's Theorem* (CHT) (Harary, 1953). Next, we construct an *ideal low-pass (LP) filter*—parameterized by the lone cutoff frequency $\omega$—for the corresponding positive graph $\mathcal{G}^+$ to minimize a denoising objective. The ideal LP filter is efficiently implemented via Lanczos approximation (Susnjara et al., 2015), which we unroll into a filter sub-network. The pair of LP filter / graph learning module is repeated to build a feed-forward network for sparse parameter learning (Thuc et al., 2024; Cai et al., 2025).

Having learned two denoisers $\mathbf{\Psi}_0(\cdot)$ and $\mathbf{\Psi}_1(\cdot)$ trained on signals from two different classes 0 (healthy subjects) and 1 (epilepsy patients)—thus capturing their respective posterior probabilities—we use their reconstruction errors on an input signal for binary classification. Experiments show that our classification method based on trained balanced signed graph denoisers achieves comparable performance as SOTA DL schemes, while employing drastically fewer parameters.

Summarizing, our key contributions are as follows:

1. Extending Thuc et al. (2024) that focuses on positive graphs, we unroll a denoising algorithm for signals on *balanced signed graphs with well-defined frequencies*—learned directly from data via feature distance learning—into a lightweight and interpretable transformer.

2. We implement an ideal LP filter on the positive graph $\mathcal{G}^+$ corresponding to each learned balanced signed graph $\mathcal{G}^B$ (Dinesh et al., 2025) without eigen-decomposition in linear time via Lanczos approximation (Susnjara et al., 2015), where only the filter cutoff frequency $\omega$ requires tuning from data.

3. We train two class-specific denoisers to learn two different posterior probabilities as a pretext task, then determine class assignment based on their reconstruction errors. This approach bridges *generative modeling* and *discriminative classification* in a novel manner—both the algorithm-unrolled denoisers and the classification decision are easily interpretable.

4. Compared to SOTA DL methods, we achieve competitive classification performance on EEG signals distinguishing epilepsy patients from healthy subjects, while using significantly fewer parameters (*e.g.*, our scheme achieves $97.6\%$ classification accuracy to transformer-based model (Lih et al., 2023)'s $85.1\%$, **while employing fewer than $1\%$ of the parameters**).

## 2 PRELIMINARIES

### 2.1 GRAPH SIGNAL PROCESSING DEFINITIONS

A graph $\mathcal{G}(\mathcal{N}, \mathcal{E}, \mathbf{W})$ is defined by a node set $\mathcal{N} = \{1, \ldots, N\}$, an edge set $\mathcal{E}$, and an *adjacency matrix* $\mathbf{W} \in \mathbb{R}^{N \times N}$, where $W_{i,j} = w_{i,j}$ is the weight of edge $(i,j) \in \mathcal{E}$ if it exists, and $W_{i,j} = 0$

---

[2]See Appendix A for related works.

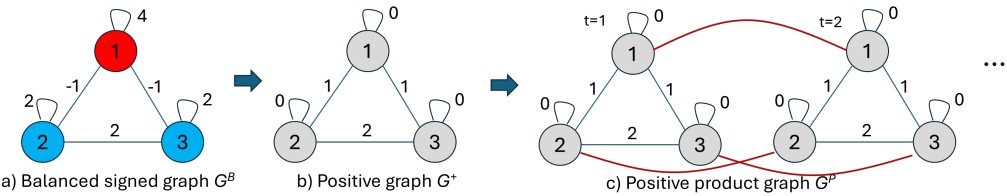

Figure 1: Example of a balanced signed graph $\mathcal{G}^B$ in (a) and its corresponding positive graph $\mathcal{G}^+$ in (b). Red and blue nodes in $\mathcal{G}^B$ denote polarities $-1$ and $1$, respectively. Positive graph can be extended to incorporate the time dimension via a product graph $\mathcal{G}^P$ in (c) with temporal edges (red).

otherwise. In this work, we assume that each edge weight $w_{i,j}$ can be positive or negative to denote positive / negative correlations; $\mathcal{G}$ with both positive and negative edges is a *signed graph*. We assume also that edges are bidirectional, and thus $w_{i,j} = w_{j,i}$ and $\mathbf{W}$ is symmetric. A *combinatorial graph Laplacian* is defined as $\mathbf{L} \triangleq \mathbf{D} - \mathbf{W} = \mathrm{diag}(\mathbf{W1}) - \mathbf{W}$. To account for self-loops, *i.e.*, $\exists i, W_{i,i} \neq 0$, a *generalized graph Laplacian* is typically used: $\mathcal{L} \triangleq \mathbf{D} - \mathbf{W} + \mathrm{diag}(\mathbf{W})$ (Ortega et al., 2018). We use these Laplacian definitions for both positive and signed graphs.

## 2.2 Graph Laplacian Regularizer

To quantify variation of a signal $\mathbf{x}$ over a graph kernel $\mathcal{G}$, the *graph Laplacian regularizer* (GLR) (Pang & Cheung, 2017) is commonly used, defined using combinatorial Laplacian $\mathbf{L}$ as

$$\mathbf{x}^\top \mathbf{L} \mathbf{x} = \sum_{(i,j)\in\mathcal{E}} w_{i,j}(x_i - x_j)^2. \tag{1}$$

From eq. (1), one can see that $\mathbf{x}^\top \mathbf{L} \mathbf{x} \geq 0, \forall \mathbf{x}$ ($\mathbf{L}$ is *positive semi-definite* (PSD)) if $\mathbf{L}$ specifies a positive graph $\mathcal{G}^+$, *i.e.*, $w_{i,j} \geq 0, \forall i, j$.

## 2.3 Balanced Signed Graphs

A *balanced signed graph*, denoted by $\mathcal{G}^B$, is a graph with no cycle of odd number of negative edges. An equivalent definition of graph balance is through *node polarities*. Each node $i \in \mathcal{V}$ is first assigned a polarity $\beta_i \in \{1, -1\}$. By the *Cartwright-Harary's Theorem* (CHT) (Harary, 1953), a signed graph is *balanced* iff positive/negative edges always connect node-pairs of the same/opposite polarities. In mathematical terms, a signed graph $\mathcal{G}$ is balanced if

$$\beta_i \beta_j = \mathrm{sign}(w_{i,j}), \qquad \forall (i,j) \in \mathcal{E}. \tag{2}$$

Recently, Dinesh et al. (2025) proved that there exists a simple similarity transform from the (generalized) graph Laplacian $\mathcal{L}^B$ of a balanced signed graph $\mathcal{G}^B$ to a graph Laplacian $\mathcal{L}^+$ of a corresponding positive graph $\mathcal{G}^+$, *i.e.*,

$$\mathcal{L}^+ = \mathbf{T} \mathcal{L}^B \mathbf{T}^{-1}, \tag{3}$$

where $\mathbf{T} = \mathrm{diag}(\boldsymbol{\beta})$ is a diagonal matrix with diagonal entries equal to node polarities $\boldsymbol{\beta} = [\beta_1, \ldots, \beta_N]$ in $\mathcal{G}^B$. Thus, $\mathcal{L}^B$ and $\mathcal{L}^+$ share the same eigenvalues, while $\mathcal{L}^B$'s eigenvectors $\mathbf{V}^B = \mathbf{T}\mathbf{V}^+$ are a linear transform of $\mathcal{L}^+$'s eigenvectors $\mathbf{V}^+$. As an example, consider the 3-node balanced signed graph $\mathcal{G}^B$ in Fig. 1 (a) and its corresponding positive graph $\mathcal{G}^+$ in (b). $\mathcal{G}^B$ is balanced since positive/negative edges connect node-pairs of same/opposite polarities. The balanced signed graph Laplacian $\mathcal{L}^B$ and the corresponding positive graph Laplacian $\mathcal{L}^+$ are

$$\mathcal{L}^B = \begin{bmatrix} 2 & 1 & 1 \\ 1 & 3 & -2 \\ 1 & -2 & 3 \end{bmatrix}, \quad \mathcal{L}^+ = \begin{bmatrix} 2 & -1 & -1 \\ -1 & 3 & -2 \\ -1 & -2 & 3 \end{bmatrix}, \tag{4}$$

where $\mathbf{T} = \mathrm{diag}([-1\ 1\ 1])$. Given that the graph frequencies of positive graphs are well established[3] (Ortega et al., 2018), the graph frequencies of balanced signed graphs are also rigorously defined.

---

[3]Specifically, eigenvectors of a positive graph Laplacian for increasing eigenvalues have non-decreasing numbers of *nodal domains* that quantify signal variation across the graph kernel (Davies et al., 2000), and hence can be rightfully interpreted as frequency components (*Fourier modes*). See Dinesh et al. (2025) for details.

## 3 BALANCED SIGNED GRAPH CONSTRUCTION & SIGNAL DENOISING

We first discuss construction of a balanced signed graph $\mathcal{G}^B$ in Section 3.1, which is mapped to a positive graph $\mathcal{G}^+$ via similarity transform of Laplacian matrices. We describe a denoiser $\Psi(\cdot)$ for signals on $\mathcal{G}^+$ based on ideal LP filtering in Section 3.2. Finally, we discuss how Lanczos approximation is used to efficiently implement a LP filter in linear time.

### 3.1 BALANCED SIGNED GRAPH CONSTRUCTION

**Polarity Selection:** We construct a balanced signed graph $\mathcal{G}^B$ to connect nodes representing EEG sensors in an electrode array of size $N$. Typically, collected data at a sensor $i \in \mathcal{V}$ is a time-series signal $x_i[n], n \in \mathbb{Z}_+$. We divide it into $H$ *chunks* of duration $D$ each, and consecutive chunks of the same sensor are connected in time using positive edges in a *product graph* $\mathcal{G}^P$ of $N \times H$ nodes, as shown in Fig. 1(c). For simplicity, we assume a single chunk in the sequel, focusing on $\mathcal{G}^B$.

To ensure balance in $\mathcal{G}^B$, we first initialize polarity $\beta_i$ for each node $i$ as follows. Given an empirical covariance matrix $\bar{\mathbf{C}} \in \mathbb{R}^{N \times N}$ computed from collected EEG data, we select one row $i$ and initialize node $i$'s polarity $\beta_i \leftarrow 1$. Then, for each $j, j \neq i$, we initialize $\beta_j \leftarrow \text{sign}(\bar{C}_{i,j})$, *i.e.*, node $j$ has the same polarity as node $i$ if $\bar{C}_{i,j} > 0$ (positively correlated), and opposite polarity otherwise.

At each subsequent graph learning module (see Section 4.1), polarities $\beta_i$'s are updated. Given a set of computed edge weights $\{w_{i,j}\}$, for each node $i$, we first assume a polarity $\beta_i \leftarrow \{1, -1\}$ and flip signs of $\{w_{i,j}\}_{j \neq i}$ so that the graph balance condition eq. (2) is satisfied, resulting in balanced signed graph Laplacian $\mathbf{L}^B(\beta_i)$. Using a set of training signals $\{\mathbf{x}^q\}_{q=1}^{Q}, \mathbf{x}^q \in \mathbb{R}^N$, we select polarity $\beta_i^*$ for node $i$ with the smaller GLR term eq. (1):

$$\beta_i^* = \arg \min_{\beta_i \in \{1,-1\}} \sum_{q=1}^{Q} (\mathbf{x}^q)^\top \mathbf{L}^B(\beta_i) \mathbf{x}^q. \tag{5}$$

In words, eq. (5) chooses polarity $\beta_i^*$ that results in a graph $\mathcal{G}^B$ more consistent / smooth with dataset $\{\mathbf{x}^q\}_{q=1}^{Q}$, similar in concept as previous works that learn graph Laplacians from assumed smooth signals (Dong et al., 2016; Kalofolias, 2016; Dong et al., 2019).

We update each node $i$'s polarity and corresponding edge weight signs in turn until convergence. The analysis of initialization statistics and the corresponding setup, including the update order, invocation strategy, and stopping criterion, are provided in detail in Appendix J, along with the pseudocode.

**Feature Distance**: Given polarities $\{\beta_i\}$, we compute signed edge weights $\{w_{i,j}\}$. For each node $i$, we assume that a *feature function* $F : \mathbb{R}^E \mapsto \mathbb{R}^K$ (to be detailed in Section 4.1) computes a low-dimensional representative *feature vector* $\mathbf{f}_i = F(\mathbf{e}_i), \mathbf{f}_i \in \mathbb{R}^K$, from *input embedding* $\mathbf{e}_i \in \mathbb{R}^E$, where $K \ll E$. Given $\mathbf{f}_i$'s, the *Mahalanobis distance* between nodes $i$ and $j$ is computed as

$$d_{i,j} = (\mathbf{f}_i - \mathbf{f}_j)^\top \mathbf{M}(\mathbf{f}_i - \mathbf{f}_j), \tag{6}$$

where $\mathbf{M} \in \mathbb{R}^{K \times K}$ is a PSD *metric matrix*, so that $d_{i,j} \geq 0, \forall \mathbf{f}_i, \mathbf{f}_j$ (Yang et al., 2022).

For each edge $(i,j) \in \mathcal{E}$, we compute signed edge weight $w_{i,j}$ as

$$w_{i,j} = \begin{cases} \exp(-d_{i,j}) & \text{if } \beta_i = \beta_j \\ \exp(-d_{i,j}) - 1 & \text{o.w.} \end{cases}. \tag{7}$$

We see that $w_{i,j} \geq 0$ ($w_{i,j} \leq 0$) if nodes $i$ and $j$ have the same (opposite) polarities; thus, by eq. (2), eq. (7) ensures the constructed signed graph $\mathcal{G}^B$ is balanced. In either case, larger feature distance $d_{i,j}$ means smaller edge weight $w_{i,j}$. Note that we are the first to map non-negative learned feature distances $d_{i,j}$'s to *signed* edge weights $w_{i,j}$'s of a balanced signed graph.

**Normalization**: We perform the following normalization for weight $w_{i,j}$ of each edge $(i,j) \in \mathcal{E}$:

$$\bar{w}_{i,j} = \frac{w_{i,j}}{\sqrt{\sum_{l \,|\, (i,l) \in \mathcal{E}} |w_{i,l}|}\sqrt{\sum_{k \,|\, (k,j) \in \mathcal{E}} |w_{k,j}|}} = \frac{\beta_i \beta_j \, \exp(-d_{i,j})}{\sqrt{\sum_{l \,|\, (i,l) \in \mathcal{E}} \exp(-d_{i,l})}\sqrt{\sum_{k \,|\, (k,j) \in \mathcal{E}} \exp(-d_{k,j})}}. \tag{8}$$

The resulting adjacency matrix $\bar{\mathbf{W}}^B$ eq. (8) is a symmetric normalized variant of $\mathbf{W}^B$.

**PSDness:** Combinatorial graph Laplacian[4] $\bar{\mathbf{L}}^B = \bar{\mathbf{D}}^B - \bar{\mathbf{W}}^B$ may not be PSD due to the presence of negative edges. To ensure PSDness, we leverage the *Gershgorin Circle Theorem* (GCT) (Varga, 2004) and add a *self-loop* of weight $\bar{w}_{i,i} = \delta$ to each node $i$, where $\delta$ is computed as

$$\lambda_{\min}^- = \min_i \bar{L}_{i,i}^B - \sum_{j|j\neq i} |\bar{L}_{i,j}^B|, \qquad \delta = \max\left(-\lambda_{\min}^-, 0\right). \tag{9}$$

$\lambda_{\min}^-$ in eq. (9) is a lower bound of the smallest eigenvalue $\lambda_{\min}$ of $\bar{\mathbf{L}}^B$ by GCT: each eigenvalue $\lambda$ of a symmetric real matrix $\mathbf{P}$ must reside inside at least one Gershgorin disc $i$ with center $center_i = P_{i,i}$ and radius $r_i = \sum_{j|j\neq i} |P_{i,j}|$, *i.e.*, $\exists i$ such that $center_i - r_i \leq \lambda \leq center_i + r_i$. A corollary is that the smallest Gershgorin disc left-end—$\lambda_{\min}^-$ in eq. (9)—is a lower bound for $\lambda_{\min}$. Thus, eq. (9) implies that the eigenvalues of $\bar{\mathbf{L}}^B$ are shifted up by $\delta$ via $\mathcal{L}^B = \bar{\mathbf{L}}^B + \delta\mathbf{I}$ to ensure $\mathcal{L}^B$ is PSD if $\lambda_{\min}^- < 0$. Note that $\mathcal{L}^B = \bar{\mathbf{L}}^B + \delta\mathbf{I}$ and $\bar{\mathbf{L}}^B$ share the same eigenvectors, and thus the self-loop additions do not affect the spectral content of $\bar{\mathbf{L}}^B$.

## 3.2 GRAPH SIGNAL DENOISING

We construct a signal denoiser, given an underlying balanced signed graph $\mathcal{G}^B$ specified by graph Laplacian $\mathcal{L}^B$. To process signals on a more convenient positive graph $\mathcal{G}^+$, we first perform a similarity transform to obtain its corresponding positive graph Laplacian, $\mathcal{L}^+ = \mathbf{T}\mathcal{L}^B\mathbf{T}^{-1}$ in eq. (3), where $\mathbf{T} = \text{diag}(\boldsymbol{\beta})$. We employ the graph spectrum of $\mathcal{L}^+$ for ideal LP filtering. Each target signal $\mathbf{y}^B$ on $\mathcal{G}^B$ to be denoised is also pre-processed to $\mathbf{y}^+ = \mathbf{T}\mathbf{y}^B$ as a signal on $\mathcal{G}^+$.

Denote by $\mathcal{S}_\omega(\mathcal{L}^+)$ the low-frequency subspace spanned by the first $\omega$ eigenvectors (frequency components) $\mathbf{V}_\omega = [\mathbf{v}_1; \mathbf{v}_2; \ldots; \mathbf{v}_\omega] \in \mathbb{R}^{N\times\omega}$ of $\mathcal{L}^+$ corresponding to the $\omega$ smallest eigenvalues. To denoise observation $\mathbf{y}^+ \in \mathbb{R}^N$, we seek a signal $\mathbf{x} \in \mathbb{R}^N$ in $\mathcal{S}_\omega(\mathcal{L}^+)$ closest to $\mathbf{y}^+$ in $\ell_2$-norm[5]:

$$\min_{\mathbf{x}\in\mathcal{S}_\omega(\mathcal{L}^+)} \|\mathbf{y}^+ - \mathbf{x}\|_2^2. \tag{10}$$

Denote by $\mathbf{z} \in \mathbb{R}^\omega$ the $\omega$ GFT coefficients of $\mathbf{x}$, *i.e.*, $\mathbf{x} = \mathbf{V}_\omega\mathbf{z}$. The optimal solution $\mathbf{z}^*$ to eq. (10) is

$$\mathbf{z}^* = (\mathbf{V}_\omega^\top\mathbf{V}_\omega)^{-1}\mathbf{V}_\omega^\top\mathbf{y}^+ \overset{(a)}{=} \mathbf{V}_\omega^\top\mathbf{y}^+ \tag{11}$$

$$\mathbf{x}^* = \mathbf{V}_\omega\mathbf{z}^* = \mathbf{V}_\omega\mathbf{V}_\omega^\top\mathbf{y}^+ = \underbrace{\mathbf{V}g_\omega(\boldsymbol{\Lambda})\mathbf{V}^\top}_{g_\omega(\mathcal{L}^+)}\mathbf{y}^+ \tag{12}$$

where $(a)$ is true since columns of $\mathbf{V}$ are orthonormal by the Spectral Theorem (Hawkins, 1975). $g_\omega(\mathcal{L}^+) = \mathbf{V}g_\omega(\boldsymbol{\Lambda})\mathbf{V}^\top$ is an *ideal LP filter*, and $g_\omega(\boldsymbol{\Lambda}) = \text{diag}([g_\omega(\lambda_1), \ldots, g_\omega(\lambda_N)])$ has *frequency response* $g_\omega(\lambda_i)$ defined as

$$g_\omega(\lambda_i) = \begin{cases} 1 & \text{if } i \leq \omega \\ 0 & \text{o.w.} \end{cases} . \tag{13}$$

Solution $\mathbf{x}^*$ in eq. (12) is an *orthogonal projection* of input $\mathbf{y}^+$ onto $\mathcal{S}_\omega(\mathcal{L}^+)$. Computing $\mathbf{x}^*$ in eq. (12) requires computation of the first $\omega$ eigenvectors $\mathbf{V}_\omega$ of $\mathcal{L}^+$ with complexity $\mathcal{O}(N^3)$ for $\omega \approx N$. In implementation, the cutoff frequency $\omega$ in eq. (13) is made learnable by approximating the hard truncation with a sigmoid-based smooth low pass filter. Detailed formulations are provided in Appendix E.3.

**Lanczos Low-pass Filter Approximation**

Instead of an ideal LP filter in eq. (13), we approximate it via Lanczos approximation in complexity $\mathcal{O}(N)$ (Susnjara et al., 2015). In a nutshell, instead of eigen-decomposing a large matrix $\mathcal{L}_m^+ \in \mathbb{R}^{N\times N}$, via the Lanczos method we operate on a much smaller tri-diagonal matrix $\mathbf{H}_m \in \mathbb{R}^{m\times m}$, where $m \ll N$ is the dimension of the approximating Krylov space. We eigen-decompose $\mathbf{H}_m = \mathbf{Z}_m g(\boldsymbol{\Lambda}_m)\mathbf{Z}_m^\top$, where $g_\xi(\lambda_i)$ is the approximate LP frequency response for cutoff frequency $\xi = \text{round}(\frac{\omega m}{N})$, which we tune from data after unrolling. See Appendix B for details.

---

[4]A signed graph Laplacian $\mathbf{L}^s \triangleq \mathbf{D}^s - \bar{\mathbf{W}}^B$, where $D_{i,i}^s = \sum_j |\bar{W}_{i,j}^B|$, guaranteed to be PSD can be defined instead (Dittrich & Matz, 2020), but a corresponding LP filter would promote *negative linear dependence* rather than *repulsions* for negative edges during signal reconstruction. See Appendix C for details.

[5]An alternative is a *maximum a posteriori* (MAP) denoising formulation using GLR as a signal prior (Pang & Cheung, 2017; Zeng et al., 2020; Dinesh et al., 2020), *i.e.*, $\min_{\mathbf{x}} \|\mathbf{y}^+ - \mathbf{x}\|_2^2 + \mu\,\mathbf{x}^\top\mathcal{L}^B\mathbf{x}$. However, Bai et al. (2020) shows that the MAP problem—called the *E-optimality criterion* in optimal design—minimizes the worst-case signal reconstruction, while eq. (10) is the *A-optimality criterion* that minimizes the average case.

Figure 2: Unrolled Graph Signal Denoising Network. Low-pass filter (LPF) computing a solution $\mathbf{x}^*$ via eq. (12) is interleaved with a balanced graph learning (BGL) module that updates balanced signed graph Laplacian $\mathcal{L}^B$, then transforms to $\mathcal{L}^+$ via eq. (3). $\boldsymbol{\Theta}_t$ and $\boldsymbol{\Phi}_t$ are learned parameters.

## 4 ALGORITHM UNROLLING

We implement the graph-based denoising procedure in eq. (12) and a graph learning module repeatedly; after a solution $\mathbf{x}^*$ is obtained, representative features $\{\mathbf{f}_i\}$ are updated (see Section 4.1), resulting in new feature distances $\{d_{i,j}\}$ via eq. (6), new signed edge weights $\{w_{i,j}\}$ via eq. (7), and new balanced signed graph Laplacian $\mathcal{L}^B$ and positive graph Laplacian $\mathcal{L}^+$ via similarity transform eq. (3). The concept of iteratively filtering signals, with filter weights updated based on computed signals, is analogous to *bilateral filter* (BF) in image denoising (Tomasi & Manduchi, 1998).

### 4.1 GRAPH LEARNING MODULE

We unroll this repeated combo of low-pass filter / graph learning module into neural layers to compose a feed-forward network for data-driven parameter learning via back-propagation; see Fig. 2 for an illustration. The key to our unrolled neural net is the periodic insertion of a graph learning module BGL that updates Laplacian $\mathcal{L}^+$ for the next LP filter module LPF. Specifically, to compute feature vector $\mathbf{f}_i \in \mathbb{R}^K$ for each node $i$, we define input embedding $\mathbf{e}_i \in \mathbb{R}^E$ as the recovered time series signal at node $i$, and implement a shallow CNN to compute $\mathbf{f}_i = \text{CNN}(\mathbf{e}_i)$, where $K \ll E$. Metric matrix $\mathbf{M} \in \mathbb{R}^{K \times K}$ in eq. (6) is also optimally tuned. Together, the CNN parameters and $\mathbf{M}$ are the learned parameters $\boldsymbol{\Phi}_\tau$ for an unrolled block $\text{BGL}_\tau$. On the other hand, the optimal cutoff frequency $\omega$ for the low-pass filter is learned per block, which constitutes parameters $\boldsymbol{\Theta}_\tau$ for $\text{LPF}_\tau$.

### 4.2 SELF-ATTENTION MECHANISM

We review the classical self-attention mechanism in transformers (Vaswani et al., 2017). First, given *input embedding* $\mathbf{e}_i \in \mathbb{R}^E$ for token $i$, *affinity* $e_{i,j}$ between tokens $i$ and $j$ is computed as the scaled product of $\mathbf{K}\mathbf{x}_i$ and $\mathbf{Q}\mathbf{x}_j$, where $\mathbf{K}, \mathbf{Q} \in \mathbb{R}^{E \times E}$ are the *key* and *query* matrices, respectively. Using $e_{i,j}$, non-negative and normalized *attention weights* $a_{i,j}$'s are computed using the softmax operator:

$$a_{i,j} = \frac{\exp(e_{i,j})}{\sum_k \exp(e_{i,k})}, \qquad e_{i,j} = (\mathbf{Q}\mathbf{e}_j)^\top (\mathbf{K}\mathbf{e}_i). \tag{14}$$

Finally, output embedding $\mathbf{y}_i$ is computed as the sum of attention-weighted input embeddings multiplied by the *value* matrix $\mathbf{V} \in \mathbb{R}^{E \times E}$:

$$\mathbf{y}_i = \sum_j a_{i,j} \mathbf{e}_i \mathbf{V}. \tag{15}$$

A transformer concatenates self-attention operations both in series and in parallel (called multi-head).

**Remark:** Comparing eq. (14) to the right-hand side of eq. (8), we see that by interpreting negative distance $-d_{i,j}$ as affinity $e_{i,j}$, normalized edge weights $\bar{w}_{i,j}$ are essentially attention weights $a_{i,j}$. Thus, *a graph learning module with normalized edge weights is a form of self-attention.* In implementation, instead of learning dense and large key and query matrices $\mathbf{K}$ and $\mathbf{Q}$, for normalized edge weights $\{\bar{w}_{i,j}\}$ we learn only parameters for a shallow CNN to compute features $\{\mathbf{f}_i\}$ and low-dimensional metric matrix $\mathbf{M}$. Further, instead of learning dense and large value matrix $\mathbf{V}$, we learn a single cutoff frequency $\omega$ of an ideal LP filter per block. Thus, our graph-based implementation of self-attention yields substantial parameter savings compared to the classical self-attention mechanism.

## 5 USING GRAPH-BASED DENOISERS FOR CLASSIFICATION

By training two class-conditioned denoisers, $\boldsymbol{\Psi}_0(\cdot)$ and $\boldsymbol{\Psi}_1(\cdot)$, using a squared-error loss function on signal classes corresponding to healthy subjects and epilepsy patients, respectively, we are training each network to compute the *posterior mean* of its corresponding class (though the networks are inspired by subspace projection); *i.e.*, given noisy signal $\mathbf{y}$ and known class $c$, they compute

$$\boldsymbol{\Psi}_0(\mathbf{y}) \approx \mathbb{E}[\mathbf{x} \mid \mathbf{y}, c = 0], \qquad \boldsymbol{\Psi}_1(\mathbf{y}) \approx \mathbb{E}[\mathbf{x} \mid \mathbf{y}, c = 1]. \tag{16}$$

To accomplish eq. (16), the two denoisers must learn *implicitly* the posterior probabilities of the two classes. By the Bayes Theorem, the posterior probability $\Pr(\mathbf{x}|\mathbf{y}, c)$ of signal $\mathbf{x}$ given observation $\mathbf{y}$ is proportional to the product of likelihood $\Pr(\mathbf{y}|\mathbf{x}, c)$ times prior $\Pr(\mathbf{x}|c)$:

$$\Pr(\mathbf{x}|\mathbf{y}, c) \propto \Pr(\mathbf{y}|\mathbf{x}, c) \Pr(\mathbf{x}|c). \tag{17}$$

Assuming zero-mean *additive white Gaussian noise* (AWGN) with variance $\sigma_n^2$, the likelihood is

$$\Pr(\mathbf{y}|\mathbf{x}, c) = \frac{1}{(2\pi\sigma_n^2)^{N/2}} \exp\left(-\frac{\|\mathbf{y} - \mathbf{x}\|_2^2}{2\sigma_n^2}\right). \tag{18}$$

Given our assumption that signal $\mathbf{x}$ resides in low-frequency subspace $\mathcal{S}_\omega(\mathcal{L}^+)$, the prior is

$$\Pr(\mathbf{x}|c) = \begin{cases} 1 & \text{if } \mathbf{x} \in \mathcal{S}_\omega(\mathcal{L}^+) \\ 0 & \text{o.w.} \end{cases} . \tag{19}$$

Thus, given cutoff frequency $\omega$, the signal $\mathbf{x}$ that maximizes the posterior $\Pr(\mathbf{x}|\mathbf{y}, c)$ is the signal $\mathbf{x}^*$ in $\mathcal{S}_\omega(\mathcal{L}^+)$ closest in Euclidean distance to $\mathbf{y}$ (so that $\Pr(\mathbf{x}|c) > 0$ and $\Pr(\mathbf{y}|\mathbf{x}, c)$ is maximized):

$$\mathbf{x}^* = \arg \min_{\mathbf{x} \in \mathcal{S}_\omega(\mathcal{L}^+)} \|\mathbf{y} - \mathbf{x}\|_2^2, \tag{20}$$

*i.e.*, the orthogonal projection of $\mathbf{y}$ onto $\mathcal{S}_\omega(\mathcal{L}^+)$ in eq. (12).

Conversely, during supervised training of a denoiser $\boldsymbol{\Psi}_c(\cdot)$, given training pairs $\{(\mathbf{y}_{c,i}, \mathbf{x}_{c,i})\}$ of class $c$, parameter set $\boldsymbol{\Phi}$ (CNN parameters and cutoff frequency $\omega$) is tuned to minimize the sum of distances between ground truths $\mathbf{x}_{c,i}$ and projections $g_\omega(\mathcal{L}^+)$ of inputs $\mathbf{y}_{c,i}$ onto $\mathcal{S}_\omega(\mathcal{L}^+)$:

$$\min_{\boldsymbol{\Phi}} \sum_i \|\mathbf{x}_{c,i} - g_\omega(\mathcal{L}^+)\mathbf{y}_{c,i}\|_2^2. \tag{21}$$

Thus, learning of parameter set $\boldsymbol{\Phi}$ at different layers in our unrolled network to minimize eq. (21) amounts to learning of posterior $\Pr(\mathbf{x}|\mathbf{y}, c)$. (Note that noisy signals $\mathbf{y}_i$'s with non-negligible noise variance $\sigma_n^2$ are necessary; a noiseless signal $\mathbf{y}_i = \mathbf{x}_i$ means that setting $\omega \leftarrow N$—resulting in an *all-pass* filter—would yield zero error in eq. (21), and thus no learning of posterior $\Pr(\mathbf{x}|\mathbf{y}, c)$.)

Once the two denoisers are trained, we compute the following given input signal $\mathbf{y}$ to determine $\mathbf{y}$'s class membership:

$$c^* = \arg \min_{c \in \{0,1\}} \|\mathbf{y} - \boldsymbol{\Psi}_c(\mathbf{y})\|_2^2. \tag{22}$$

The reasoning is as follows: given that denoiser $\boldsymbol{\Psi}_c(\mathbf{y})$ computes the posterior mean $\mathbb{E}[\mathbf{x} \mid \mathbf{y}, c]$ which is the *minimum mean square error* (MMSE) estimator, its error should be small when $\mathbf{y}$ indeed belongs to class $c$. Hence, classification by reconstruction errors[6] in eq. (22) is justified.

**Modified Training Objective:** To encourage discrimination of the two classes, we adopt a new loss function during denoiser training. We first identify pairs of signals $(\mathbf{x}_{0,i}, \mathbf{x}_{1,i})$ from the two classes that are close in Euclidean distance (and thus difficult to differentiate). We then train denoiser $\boldsymbol{\Psi}_0(\cdot)$ for class 0 with the following *contrastive loss function* (similar training procedure for $\boldsymbol{\Psi}_1(\cdot)$):

$$\sum_i \|\mathbf{x}_{0,i} - \boldsymbol{\Psi}_0(\mathbf{y}_{0,i})\|_2^2 + \max\left(\rho - \|\mathbf{x}_{1,i} - \boldsymbol{\Psi}_0(\mathbf{y}_{1,i})\|_2^2, 0\right), \tag{23}$$

where $\rho > 0$ is a parameter, and $\mathbf{y}_{c,i}$ is a noisy version of $\mathbf{x}_{c,i}$. Doing so means that $\boldsymbol{\Psi}_0(\cdot)$ captures signal statistics for class 0 that are sufficiently different from class 1.

---

[6]It is also the maximum likelihood estimate (MLE). See Appendix D for explanation.

# 6 EXPERIMENTS

## 6.1 EXPERIMENTAL SETUP

**Datasets and settings.** We evaluate our model on the Turkish Epilepsy EEG Dataset (Tasci et al., 2023), which is currently the largest publicly available dataset focused on epileptic seizures. The dataset comprises 10,356 EEG recordings collected from 121 participants, including 50 patients diagnosed with generalized epilepsy and 71 healthy controls. Each recording contains 35 channels of EEG signals sampled at 500 Hz for a duration of 15 seconds. To mitigate artifacts typically observed at the beginning and end of recordings, we discard the first 2 seconds and the last 1 second.

As the default classification task setting, we follow (Tasci et al., 2023; Lih et al., 2023; Shen et al., 2024; Bhandage et al., 2024; Dicsli et al., 2025) and divide the dataset into training, validation, and test sets in a ratio of 8: 1: 1. To assess the model's generalization across different subjects, we also perform a leave-one-out-subject (LOSO) classification task. In this setting, data from one subject is held out as the test set, while the remaining data is used for training and validation. The training and validation sets are used for the denoising task, while the test set is reserved for classification.

For graph construction[7], each remaining 6,000-point (12 second) sequence is segmented into 6 non-overlapping chunks, resulting in a temporal graph of length 6, where each node corresponds to a 1000-dimensional feature vector. We employ three stacked blocks, each consisting of a BGL module with three convolutional layers followed by the LPF operation as shown in Figure 2. The term block here refers to this BGL + LPF unit, which is consistently used in the subsequent ablation studies. All models are trained on an NVIDIA GeForce RTX 3090.

**Baseline methods.** We compare the proposed method with several competitive baselines, including both graph-based and non-graph-based approaches. (Tasci et al., 2023) use a k-Nearest Neighbors (kNN) classifier combined with Multivariate Dynamic Time Warping (MDTW). The Transformer-based method of (Lih et al., 2023) models temporal dependencies in EEG time-series data for classification. (Shen et al., 2024) employ Regularized O-minus tensor network decomposition (ROD) to extract features from time–frequency (TF) maps. (Mohammadi Foumani et al., 2024) propose a self-supervised masked reconstruction framework to learn robust temporal–spatial EEG representations. Convolutional Neural Networks (CNNs) are used by (Bhandage et al., 2024; Dicsli et al., 2025) to extract discriminative features from EEG spectrograms, while (Pan et al., 2022) further enhance feature quality by processing channel-wise embeddings on a Riemannian manifold. On the graph-based side, DGCNN (Song et al., 2018) and GIN (Zhang & Yao, 2021) represent EEG data as graphs to capture complex inter-channel relations, while EEGNet (Lawhern et al., 2018) and FBCSPNet (Schirrmeister et al., 2017) incorporate graph-inspired operations to enhance feature extraction and classification performance.

## 6.2 EXPERIMENTAL RESULTS

### 6.2.1 MAIN RESULTS

Table 1 presents a detailed comparison of our method against several existing approaches, divided into two main categories: non-graph-based methods in the default task setting and graph-based methods in the leave-one-subject-out (LOSO) task setting.

In the default task setting, which involves training and testing on the same dataset split, our model outperforms most baselines in terms of accuracy, precision, specificity, and F1-score, achieving 97.57%, 98.58%, 97.45%, and 98.01% respectively, with only 14,787 parameters. This demonstrates the efficiency and effectiveness of our method. While (Bhandage et al., 2024) based on STFT and CNN achieves a higher accuracy of 99.20% and F1-score of 99.30%, it requires several orders of magnitude more parameters (over 11.5 million), making it computationally expensive and memory-inefficient.

In addition to the default setting, we also evaluate our model under the more challenging LOSO task setting, which tests the model's ability to generalize across different subjects. Here, our model demonstrates strong performance with 90.06% accuracy and 92.59% F1-score, outperforming other graph-based methods such as DGCNN, GIN, and EEGNet. This shows that our model not only excels

---

[7]See Appendix E for detailed configuration.

Table 1: **Comparison of Performance Among Different Methods.** We compare the performance of various non-graph and graph-based methods across several metrics. Our method, with a lightweight model, achieves state-of-the-art (SOTA) results in most of the metrics, and is comparable to larger models in terms of performance.

| Method | Params # | Accuracy (%) | Precision (%) | Recall (%) | Specificity (%) | F1-score (%) |
|---|---|---|---|---|---|---|
| **Non-graph-based Methods in Default Task Setting** | | | | | | |
| MDTW + KNN (Tasci et al., 2023) | - | 87.78 | 89.39 | 81.32 | 92.68 | 85.16 |
| TF + ROD (Shen et al., 2024) | 18,400 | 88.08 | 89.22 | 82.28 | 92.46 | 85.61 |
| mAtt (Pan et al., 2022) | 46,542 | 92.00 | 95.27 | 85.68 | 96.78 | 90.22 |
| EEG2Rep (Mohammadi Foumani et al., 2024) | 96,886 | 91.41 | 94.79 | 86.09 | 91.48 | 93.11 |
| CWT + DCNN (Dicsli et al., 2025) | 143,297 | 95.91 | 94.55 | **96.98** | 96.06 | 95.30 |
| **Ours** | **14,787** | **97.57** | **98.58** | 95.98 | **97.45** | **98.01** |
| **Large model in Default Task Setting** | | | | | | |
| Transformer (Lih et al., 2023) | 1,849,771 | 85.12 | 82.00 | 82.00 | 87.32 | 82.00 |
| STFT + CNN (Bhandage et al., 2024) | 11,533,928 | 99.20 | 99.14 | 99.46 | 98.98 | 99.30 |
| **Graph-based Methods in LOSO Task Setting** | | | | | | |
| DGCNN (Song et al., 2018) | 149,466 | 76.74 | 69.56 | 62.74 | 84.60 | 65.97 |
| GIN (Zhang & Yao, 2021) | 25,794 | 68.82 | 58.78 | 44.36 | 82.55 | 50.56 |
| EEGNet (Lawhern et al., 2018) | **9,170** | 78.78 | 81.26 | 53.25 | 93.11 | 64.34 |
| FBCSPNet (Schirrmeister et al., 2017) | 98,242 | 81.76 | 92.80 | 53.40 | **97.67** | 67.79 |
| Deep4Net (Schirrmeister et al., 2017) | 321,227 | 78.62 | 73.06 | 64.20 | 86.72 | 68.34 |
| **Ours(LOSO)** | 14,787 | **90.06** | **93.48** | **86.10** | 91.70 | **92.59** |

in a controlled, single-subject setting but also generalizes well to unseen subjects, a crucial aspect for real-world applications where the model needs to handle diverse and unknown data distributions. In this setting, our model maintains a good balance between computational efficiency and high generalization capability, outperforming many larger models such as Deep4Net and FBCSPNet.

By constructing an interpretable, lightweight transformer through the unrolling of graph-based algorithms, we focus on learning only the identified unknown parameters outside the optimization framework, which is specifically tailored for balanced signed graph signal denoising. This approach results in a significant reduction in the number of parameters compared to conventional 'black-box' neural network models, which are often generic and lack interpretability.

### 6.2.2 ABLATION STUDIES

We evaluate the impact of graph type on classification performance by comparing the proposed balanced signed graph with two alternatives: a positive graph and an unbalanced signed graph. The positive graph assigns all edge weights as positive, disregarding pairwise anti-correlations in data, while the unbalanced signed graph models pairwise anti-correlations using negative edges, but does not ensure graph balance, and thus graph frequencies are ill-defined. See Appendix C for details. As shown in Table 2, the balanced signed graph outperforms both alternatives, which highlights the importance of both signed edges and graph balance when modeling EEG signals and implementing LP graph filters for denoising.

In our ablation study on loss function design, we observe that using a standalone MSE loss for learning signal priors yields substantially weaker downstream classification performance compared with the contrastive MSE formulation defined in eq. (23). As shown in Table 3, incorporating contrastive guidance consistently enhances all evaluation metrics, indicating that the margin-based negative-sample penalty helps the denoiser preserve class-discriminative structure that is critical for EEG classification.

Table 2: **Ablation Study on Different Graph Types.** We compare the performance of three graph types on the LOSO task. The results highlight the importance of signed edges and graph balance.

| Setting | Accuracy (%) | Precision (%) | Recall (%) | Specificity (%) | F1-score (%) |
|---|---|---|---|---|---|
| Positive Graph | 84.30 | 88.49 | 76.88 | 86.00 | 87.23 |
| Unbalanced Signed Graph | 78.87 | 86.74 | 68.22 | 78.69 | 82.52 |
| **Balanced Signed Graph** | **93.68** | **96.32** | **89.45** | **93.61** | **94.94** |

Table 3: **Classification Results with Denoisers Trained Using Different Loss Functions.** We compare the classification performance of denoisers trained with a standalone MSE loss and a contrastive MSE loss. Incorporating contrastive guidance significantly improves all evaluation metrics.

| Loss Function | Accuracy (%) | Precision (%) | Recall (%) | Specificity (%) | F1-score (%) |
|---|---|---|---|---|---|
| Single MSE | 81.44 | 77.36 | 97.67 | 99.32 | 86.97 |
| **Contrastive MSE** | **97.57** | **98.58** | 95.98 | **97.45** | **98.01** |

To assess whether enforcing graph balance during polarity optimization may inadvertently remove meaningful anti-correlations, we examined the signed graphs constructed directly from the empirical EEG covariance matrices. Each signal was segmented into six temporal chunks, forming a product-graph structure identical to the Positive Product Graph in Fig. 1 of the main text. Edge signs were assigned based on covariance values, with a threshold of $-0.1$ marking strongly negative relations. Across the resulting signed temporal graph, we observed only 6 odd negative basic cycles out of 1481 total cycles (a fraction of 0.004), and a greedy approximation (Fontan et al., 2024) of the frustration index yielded merely 16 frustrated edges among 1322 edges. These metrics indicate that the empirical EEG graph is already very close to a perfectly balanced signed structure, with sparse and highly consistent anti-correlations. Consequently, enforcing balance makes only minimal adjustments while preserving the substantive anti-correlation patterns present in the data.

We also conducted further studies to assess the robustness and generalizability of our method. These include ablation studies on model architecture, and signal feature distance computation (Appendix F), validation on the TUH Abnormal EEG Corpus (Obeid & Picone, 2016) (Appendix G), statistical significance tests for the default classification task and the LOSO task (Appendix H), and a comparison of training and inference time for graph-based baselines (Appendix I).

## 7 CONCLUSION

To differentiate between EEG brain signals from epilepsy patients and those from healthy subjects, we unroll iterations of a balanced signed graph algorithm that minimizes a signal denoising objective into a lightweight and interpretable neural net. A balanced signed graph can capture pairwise anti-correlations in data, while retaining the frequency notion for efficient spectral filtering. Via a signed edge weight assignment that leverages the Cartwright-Harary Theorem, graph balance is ensured when mapping from learned positive feature distances. Denoising is achieved via a sequence of graph learning / ideal low-pass filtering modules, where the cutoff frequencies are learned from data. We show that our graph learning module with normalization plays the role of self-attention, and thus our graph-based denoisers are transformers. Using two denoisers trained to learn posterior probabilities of two signal classes, our method achieves competitive binary classification as SOTA deep learning models, while requiring far fewer parameters. One limitation is that our method is currently suitable only for binary classification. For future work, we consider an extension to build a multi-class classification tree from graph-based denoisers.

## 8 ACKNOWLEDGEMENT

This work was supported in part by the Beijing Major Science and Technology Project under Contract no. Z251100008125007. The work of G. Cheung was supported by the Natural Sciences and Engineering Research Council of Canada (NSERC) RGPIN-2025-06252.

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

- APPENDIX -

## LIGHTWEIGHT TRANSFORMER FOR EEG CLASSIFICATION VIA BALANCED SIGNED GRAPH ALGORITHM UNROLLING

## A    RELATED WORK IN DENOISERS AS PRETEXT TASK

Given that a denoiser can learn compact representations from sufficient training data, there are existing works that train denoisers as a pretext task for other downstream applications (Ho et al., 2020; Wu et al., 2023; Clark & Jaini, 2023). *Denoiser Diffusion Probabilistic Model* (DDPM) (Ho et al., 2020) employs a learned denoiser in a reverse path to gradually remove Gaussian noise from a pure noise image, in order to generate a realistic image. Wu et al. (2023) employs a denoising masked autoencoder to learn latent representations from Gaussian-noise-corrupted images, which can benefit downstream tasks such as classification. Clark & Jaini (2023) shows that a denoiser-based diffusion model can be repurposed for zero-shot classification. Our approach to binary EEG classification differs in the following aspects. First, we train one class-specific denoiser $\mathbf{\Psi}_c(\cdot)$ per class $c$, so that the posterior probability distribution unique to that class is learned. Second, we use reconstruction errors of the two trained denoisers operating on an input signal to determine its class assignment. In so doing, we achieve model interpretability for both the denoising step and the classification step (the denoiser is built by unrolling a graph-based denoising algorithm), while minimizing the number of parameters used.

## B    LANCZOS LOW-PASS FILTER APPROXIMATION

Similarly done in Vu et al. (2021), we approximate a low-pass graph filter output $g(\mathcal{L}^+)\mathbf{y}^+$ via Lanczos approximation (Susnjara et al., 2015) as follows. Denote by $\mathbf{U}_m \in \mathbb{R}^{m \times N}$, $m < N$, a matrix containing as columns $m$ orthonormal basis vectors of a Krylov space $\mathcal{K}_m(\mathcal{L}^+, \mathbf{y}) = \text{span}\{\mathbf{y}, \mathcal{L}^+\mathbf{y}, \dots, (\mathbf{L}^+)^{m-1}\mathbf{y}\}$. $\mathbf{U}_m$ can be computed using the Lanczos method in $\mathcal{O}(m|\mathcal{E}|)$. $\mathbf{U}_m$ tri-diagonalizes $\mathcal{L}^+ \in \mathbb{R}^{N \times N}$ into $\mathbf{H}_M \in \mathbb{R}^{m \times m}$, *i.e.*,

$$\mathbf{H}_m = \mathbf{U}_m^\top \mathcal{L}^+ \mathbf{U}_m = \begin{bmatrix} \alpha_1 & \beta_2 & & & \\ \beta_2 & \alpha_2 & \beta_3 & & \\ & \beta_3 & \alpha_3 & \ddots & \\ & & \ddots & \ddots & \beta_m \\ & & & \beta_m & \alpha_m \end{bmatrix}. \tag{24}$$

We approximate a low-pass filter $g(\mathcal{L}^+)\mathbf{y}^+$ as

$$g(\mathcal{L}^+)\mathbf{y}^+ \approx \|\mathbf{y}^+\|_2 \mathbf{U}_m g(\mathbf{H}_m)\mathbf{c}_1, \tag{25}$$

where $\mathbf{c}_1$ is the first canonical vector. Eigen-decomposition $g(\mathbf{H}_m) = \mathbf{Z}_m g(\mathbf{\Lambda}_m)\mathbf{Z}_m^\top$ can be computed in $\mathcal{O}(m^2)$ for a tridiagonal, sparse and symmetric matrix, using a specialized algorithm such as the *divide-and-conquer eigenvalue algorithm* (Cuppen, 1980). Assuming $m \ll N$ and the number of edges $|\mathcal{E}|$ is $\mathcal{O}(N)$, complexity of eq. (25) is $\mathcal{O}(N)$.

## C    USE OF CONVENTIONAL GRAPH LAPLACIAN VERSUS SIGNED GRAPH LAPLACIAN

We show that the eigenvectors of the conventional graph Laplacian $\mathbf{L} \triangleq \mathbf{D} - \mathbf{W}$ better capture pairwise (dis)similarities (quantified by feature distance in equation 6) in our signed graph $\mathcal{G}$ for LP signal reconstruction than the signed graph Laplacian $\mathbf{L}^s \triangleq \mathbf{D}^s - \mathbf{W}$, where $D_{i,i}^s = \sum_j |W_{i,j}|$ (Dittrich & Matz, 2020). Eigenvectors $\{\mathbf{v}_i\}$ of $\mathbf{L}$ are successive norm-one vectors that minimize the *Rayleigh quotient*:

$$\mathbf{v}_i = \arg \min_{\mathbf{v} \,|\, \mathbf{v} \perp \mathbf{v}_j, j < i} \mathbf{x}^\top \mathbf{L}\mathbf{x} = \sum_{(i,j) \in \mathcal{E}} w_{i,j}(x_i - x_j)^2. \tag{26}$$

For $w_{i,j} < 0$, minimizing $\mathbf{x}^\top \mathbf{L}\mathbf{x}$ promotes *repulsion*, *i.e.*, $|x_i - x_j|$ should be large, and $x_i$ and $x_j$ should be different / dissimilar.

In contrast, using the signed graph Laplacian $\mathbf{L}^s$, the Rayleigh quotient is

$$\mathbf{x}^\top \mathbf{L}^s \mathbf{x} = \sum_{(i,j)\in\mathcal{E}} |w_{i,j}|(x_i - \text{sign}(w_{i,j})x_j)^2. \tag{27}$$

If $w_{i,j} < 0$, then $|w_{i,j}|(x_i - \text{sign}(w_{i,j})x_j)^2 = |w_{i,j}|(x_i + x_j)^2$. Thus, minimizing $\mathbf{x}^\top \mathbf{L}^s \mathbf{x}$ promotes *negative linear dependence*, *i.e.*, $|x_i + x_j|$ should be small and $x_i \approx -x_j$. While negative linear dependence is a specific structured form of repulsion, they are not the same. In our case, given that negative edge weights are encoding anti-correlations, anti-correlated samples $i$ and $j$ do not imply $x_i \approx -x_j$ if they have non-zero means. Experimentally, we found that using the conventional graph Laplacian $\mathbf{L}$ to define balanced signed graph frquencies outperforms using $\mathbf{L}^s$ in EEG signal denoising and classification.

We demonstrate also the importance of signed graph edges as well as graph balance in modeling EEG data with anti-correlations. A positive graph $\mathcal{G}^+$ with positive edges can have weights defined as $w_{i,j} = \exp(-d_{i,j})$, given positive feature distance $d_{i,j}$ in eq. (6). A general signed graph $\mathcal{G}$ can define *signed* edge weight $w_{i,j} \in [-1, 1]$ using a shifted logistic function:

$$w_{i,j} = \frac{-2}{1 + e^{-(d_{i,j}-d^*)}} + 1, \tag{28}$$

where $d^* > 0$ is a parameter. Like eq. (7), eq. (28) states that edge weight $w_{i,j}$ has smaller weight for larger feature distance $d_{i,j}$ but it does not guarantee graph balance. Using the signed graph Laplacian definition $\mathbf{L}^s = \mathbf{D}^s - \mathbf{W}$, one can then perform spectral low-pass filtering as done previously. We show in Section 6 that both positive graph and unbalanced signed graph are inferior to balanced signed graph in classification performance.

## D    JUSTIFICATION FOR THE RECONSTRUCTION ERROR METRIC

We provide an alternative explanation of why the reconstruction error criterion eq. (22) to determine class assignment for an input signal $\mathbf{y}$ is reasonable. Given our assumed AWGN noise model eq. (18), the signal $\mathbf{x}^*$ that maximizes the likelihood term $\Pr(\mathbf{y} \mid \mathbf{x}, c)$ is the one between $\mathbf{x}_0^* = \mathbf{\Psi}_0(\mathbf{y})$ and $\mathbf{x}_1^* = \mathbf{\Psi}_1(\mathbf{y})$ that minimizes the numerator of the exponential function, *i.e.*,

$$\mathbf{x}^* = \arg\min_{\mathbf{x}_c^* \mid c\in\{0,1\}} \|\mathbf{y} - \mathbf{x}_c^*\|_2^2 = \arg\max_{\mathbf{x}_c^* \mid c\in\{0,1\}} \Pr(\mathbf{y} \mid \mathbf{x}_c^*, c) \tag{29}$$

which is the reconstruction error criterion. Thus, our class assignment based on reconstruction error criterion equation 22 is also the *maximum likelihood estimate* (MLE).

## E    MODEL SETUP

This section provides a detailed description of our model architecture and experimental configuration.

### E.1    GRAPH CONSTRUCTION SETUP

Since EEG signals are computed between pairs of electrodes, we model the basic spatial structure using a *line graph* derived from an undirected primary graph $\mathcal{G}^o = (\mathcal{N}^o, \mathcal{E}^o)$, where each vertex $i \in \mathcal{N}^o$ corresponds to an EEG electrode and each edge $(i, j) \in \mathcal{E}^o$ represents a bipolar EEG channel (*i.e.*, a signal computed between two electrodes). The line graph $\mathcal{G} = (\mathcal{N}, \mathcal{E})$ is then constructed such that each node $k \in \mathcal{N}$ corresponds to an edge $e_k \in \mathcal{E}^o$ in the original graph $\mathcal{G}^o$. Two nodes $k, l, \in \mathcal{N}$ in the line graph are connected by an edge $(k, l)$ in $\mathcal{E}$ if and only if the corresponding edges $e_k, e_l \in \mathcal{E}^o$ in the original graph share a common vertex. Formally,

$$\mathcal{N} = \mathcal{E}^o, \quad \mathcal{E} = \{(e_i, e_j) \in \mathcal{E}^o \times \mathcal{E}^o \mid e_i \cap e_j \neq \emptyset\}. \tag{30}$$

This construction emphasizes the edge-centric structure of EEG signal representation, which naturally aligns with the properties of bipolar recordings. To capture temporal dynamics, we segment the EEG signal of length 6000 into 6 non-overlapping temporal windows, each of length 1000. A distinct line graph is instantiated for each window, resulting in a temporal graph composed of 6 time-specific subgraphs, effectively modeling time-evolving edge dependencies.

### E.2 BALANCED GRAPH LEARNING MODULES SETUP

Each Balanced Graph Learning (BGL) module is designed to extract local temporal features from EEG edge signals using a lightweight convolutional architecture. Specifically, the module processes edge-level input through a sequence of four convolutional blocks, each consisting of a 2D convolution layer with kernel size $(1, 5)$ and stride $(1, 2)$ along the temporal dimension, followed by batch normalization and a LeakyReLU activation with a negative slope of $0.01$. This gradually compresses the temporal length while preserving the spatial (node) dimension. A final $1 \times 1$ convolution reduces the channel dimension to one, and an adaptive average pooling layer projects the output to a fixed-size feature map of shape $(N, d)$, where $N = 210$ is the number of nodes (6 time slices $\times$ 35 EEG channels) and $d = 63$ is the feature dimension per node.

To construct the graph structure within each BGL module, we compute a sample-specific signed and normalized affinity matrix $W \in \mathbb{R}^{N \times N}$ based on the extracted features $f \in \mathbb{R}^{B \times S \times N}$. A Mahalanobis-like distance is first evaluated as equation equation 6, where $\mathbf{M} = \mathbf{Q}_i \mathbf{Q}_i^\top$ with $\mathbf{Q}_i$ being a randomly initialized real matrix that is updated during training, yielding a symmetric positive semi-definite matrix $\mathbf{M}$. The distances are normalized to $[0, 1]$ per sample, and converted into affinities using a radial basis function: $w_{ij} = \exp(-d_{ij})$. These affinities are then symmetrically normalized as $\bar{\mathbf{W}} = \mathbf{D}^{-1/2} \mathbf{W} \mathbf{D}^{-1/2}$ (8) to obtain a stable and scale-invariant edge weight matrix.

### E.3 LOW-PASS FILTER MODULES SETUP

To enable learnable frequency responses in the low-pass filter modules, we adopt a parameterized sigmoid function to approximate the ideal low-pass characteristic. Specifically, given the eigenvalues $\{\lambda_i\}_{i=1}^S$ of the graph Laplacian $\mathbf{L}^+ \in \mathbb{R}^{S \times S}$, we define the frequency response function as:

$$g(\lambda_i) = \sigma \left( \alpha(\omega - \lambda_i) \right), \tag{31}$$

where $\sigma(\cdot)$ denotes the sigmoid function, $\alpha$ is a steepness parameter set to 10 controlling the sharpness of the transition band, and $\omega$ is a learnable threshold representing the cutoff frequency. which allows the model to softly suppress high-frequency components (*i.e.*, those with larger $\lambda_i$) while retaining low-frequency information in a differentiable and trainable manner. This formulation ensures smooth gradients during back-propagation and avoids the non-differentiability of hard thresholding.

### E.4 DENOISER TRAINING SETUP

All models are trained for up to 100 epochs using the Adam optimizer with an initial learning rate of $1 \times 10^{-3}$. A cosine annealing scheduler with warm restarts is applied to adjust the learning rate dynamically, with the first restart period set to $T_0 = 5$, a multiplier $T_{\text{mult}} = 1$, and a minimum learning rate of $1 \times 10^{-5}$. The parameter $\rho$ in the contrastive loss function equation 23 is fixed at 1.0. Training is conducted on a single NVIDIA GeForce 3090 GPU with a batch size of 8. Early stopping is implemented with a patience of 10 epochs based on validation performance.

## F MORE ABLATION STUDIES

In this section, we investigate the impact of different model design choices on classification performance. Specifically, we evaluate the effect of temporal sequence length, CNN block number, and distance metric selection on the overall model performance.

### F.1 ABLATION STUDY ON TEMPORAL SEQUENCE LENGTH

We explore the influence of temporal sequence length on the model's performance. As shown in Table 4, increasing the temporal sequence length from 3 to 10 consistently improves accuracy and other evaluation metrics, highlighting the model's ability to better capture temporal context. However, when the sequence length exceeds 10, performance improvements plateau, and the model's computational cost and memory consumption increase. These findings suggest diminishing returns with longer sequence lengths, emphasizing the need for a balanced choice of sequence length.

Table 4: Ablation study on the temporal sequence length

| Sequence Length | Accuracy (%) | Precision (%) | Recall (%) | Specificity (%) | F1-score (%) |
|---|---|---|---|---|---|
| 3 | 92.66 | 95.39 | 88.26 | 92.92 | 94.14 |
| 6 | 95.85 | 96.03 | 95.53 | 97.49 | 96.75 |
| 10 | 97.57 | 98.58 | 95.98 | 97.45 | 98.01 |
| 12 | 97.43 | 98.40 | 95.81 | 97.30 | 97.85 |
| 15 | 95.77 | 96.21 | 95.49 | 97.38 | 96.63 |
| 20 | 95.50 | 95.87 | 94.88 | 97.10 | 96.12 |
| 30 | 92.42 | 94.90 | 88.10 | 92.65 | 93.75 |
| 40 | 91.23 | 93.15 | 86.80 | 91.80 | 92.30 |
| 50 | 90.78 | 92.76 | 86.20 | 91.12 | 91.84 |
| 60 | 89.95 | 91.90 | 85.33 | 90.60 | 90.90 |
| 100 | 88.10 | 90.20 | 83.92 | 89.42 | 89.88 |

## F.2 ABLATION STUDY ON CNN BLOCK NUMBER

We analyze the impact of varying the number of CNN blocks on model performance. As shown in Table 5, increasing the number of CNN blocks leads to improved performance in terms of accuracy, precision, and F1-score. However, the performance improvements start to plateau after 6 blocks, with further increases in depth providing diminishing returns. This suggests that a deeper network helps improve feature extraction initially, but excessive depth increases computational complexity without significantly improving classification accuracy.

Table 5: Ablation study on the block number of CNN

| Block Numbers | Accuracy (%) | Precision (%) | Recall (%) | Specificity (%) | F1-score (%) |
|---|---|---|---|---|---|
| 1 | 65.15 | 29.12 | 92.45 | 74.50 | 41.87 |
| 3 | 94.31 | 96.57 | 90.62 | 94.37 | 95.46 |
| 4 | 96.12 | 96.00 | 94.10 | 96.20 | 95.02 |
| 6 | 97.57 | 98.58 | 95.98 | 97.45 | 98.01 |
| 9 | 97.44 | 98.40 | 96.00 | 97.40 | 97.90 |
| 12 | 97.42 | 98.35 | 95.95 | 97.38 | 97.85 |

## F.3 ABLATION STUDY ON DISTANCE METRIC

We evaluate the influence of different distance metrics on classification performance. Table 6 shows the performance of four commonly used distance metrics: Mahalanobis Distance, Euclidean Distance, Cosine Similarity, and Manhattan Distance. The Mahalanobis Distance, which incorporates trainable parameters, provides the best performance in terms of accuracy, precision, and F1-score, with an accuracy of 97.57%, precision of 98.58%, and F1-score of 98.01%. However, it requires longer convergence time (2 hours 14 minutes) compared to the simpler metrics such as Euclidean Distance, which achieves a slightly lower accuracy of 96.80% and converges in 1 hour 55 minutes. These results highlight the trade-offs between performance and computational efficiency, with Mahalanobis Distance offering the best results at the cost of increased computational overhead.

Table 6: Ablation study on the distance metric

| Distance Metric | Accuracy (%) | Precision (%) | Recall (%) | Specificity (%) | F1-score (%) | Convergence Time |
|---|---|---|---|---|---|---|
| Mahalanobis Distance | 97.57 | 98.58 | 95.98 | 97.45 | 98.01 | 2 h 14 min |
| Euclidean Distance | 96.80 | 97.85 | 94.65 | 96.55 | 96.92 | 1 h 55 min |
| Cosine Similarity | 96.40 | 97.60 | 94.80 | 96.70 | 96.90 | 1 h 40 min |
| Manhattan Distance | 96.20 | 97.35 | 94.50 | 96.40 | 96.85 | 1 h 50 min |

## G    CLASSIFICATION TASK ON TUH ABNORMAL EEG COUPUS DATASET

We also evaluated our method on the TUH Abnormal EEG Corpus (Obeid & Picone, 2016), which is widely used in epilepsy detection. This dataset consists of 2,993 EEG segments from 2,329 patients, with 70% of the data used for training and 30% for testing, following the protocol in (Chen et al., 2025). The comparison is made with several baseline results from (Chen et al., 2025). As shown in the table 7, our model achieves an accuracy of 90.69%, an F1-score of 92.60%, and a G-mean of 89.76%. These results are superior to several baseline methods. For instance, methods like BD-Deep4 and WaveNet-LSTM have lower accuracies of 85.40% and 88.76%, respectively. Traditional approaches such as DWT + CSP + CatBoost also achieve 90.22% accuracy, but our method outperforms them by achieving higher F1-scores and G-mean. Overall, our model demonstrates strong performance, surpassing a wide range of classical and deep learning methods, highlighting its effectiveness in detecting epileptic seizures in diverse datasets.

Table 7: Comparison of performance on TUH Abnormal EEG Corpus

| Method | Accuracy (%) | F1-score (%) | G-mean (%) |
|---|---|---|---|
| BD-Deep4 | 85.40 | 82.52 | 84.08 |
| AlexNet + MLP | 89.13 | 87.06 | 88.02 |
| AlexNet + SVM | 87.32 | 84.97 | 86.24 |
| WaveNet-LSTM | 88.76 | 88.32 | 88.39 |
| HT + RG | 85.86 | 83.40 | 85.19 |
| LSTM + Attention | 79.05 | 79.00 | 79.00 |
| WPD + CatBoost | 87.68 | 86.06 | 87.24 |
| Multilevel DWT + KNN | 87.68 | 86.07 | 87.24 |
| WPD + CatBoost | 89.13 | 87.60 | 88.60 |
| DWT + CSP + CatBoost | 90.22 | 88.89 | 89.76 |
| **Ours** | **90.69** | **92.60** | **89.76** |

## H    STATISTICAL SIGNIFICANCE TESTING

To ensure the robustness and statistical significance of our results, we conducted extensive experiments, including t-tests and ANOVA.

**T-test:** We conducted 100 independent experiments across 10 random data partitions, with each partition containing 10 runs, each initialized randomly. The mean classification accuracies (± standard deviation) of our method and three baseline models are presented in Table 8. Paired t-tests show that our method outperforms all baselines significantly ($p < 0.001$), demonstrating the effectiveness and robustness of our approach.

Table 8: T-test Results for Performance Comparison

| Model | Accuracy (mean $\pm$ std) | p-value vs Ours |
|---|---|---|
| Ours | $97.44 \pm 0.40$ | - |
| EEGNet | $93.30 \pm 0.60$ | $p \ll 0.0001$ |
| FBCSPNet | $96.91 \pm 0.50$ | $p\ 0.0001$ |
| Deep4Net | $96.58 \pm 0.55$ | $p\ 0.0001$ |

**ANOVA-test:** In addition to the default evaluation, we also assess cross-subject generalization on the Turkish Epilepsy EEG Dataset. Since other seizure-focused datasets utilize different electrode montages, direct cross-dataset validation is not feasible. We adopted a leave-one-subject-out (LOSO) protocol across all 121 subjects. Under this stricter data split, our model achieved 93.68% accuracy and 94.94% F1-score, demonstrating strong performance across subjects. To further evaluate the consistency of performance across individuals, we performed a one-way ANOVA on the per-subject LOSO accuracies and F1-scores. The ANOVA results, shown in Table 9, revealed no significant differences between subjects (Accuracy: $F = 0.85$, $p = 0.65$; F1-score: $F = 1.02$, $p = 0.42$), confirming stable and consistent performance across different individuals.

Table 9: ANOVA Results for Per-Subject LOSO Evaluation

| Metric | Mean $\pm$ Std (%) | ANOVA F-value | p-value |
|---|---|---|---|
| LOSO Accuracy | $93.68 \pm 1.20$ | 0.85 | 0.65 |
| LOSO F1-score | $94.94 \pm 1.10$ | 1.02 | 0.42 |

## I  COMPUTATIONAL EFFICIENCY EVALUATION

To assess the computational efficiency of our method, we compared the training and inference times of our approach with three state-of-the-art baselines: EEGNet, FBCSPNet, and Deep4Net. Our method significantly outperforms these baselines in both training and inference times. Specifically, training our model takes only 2 hours and 14 minutes, compared to 5 hours 33 minutes for EEGNet, 9 hours 1 minute for Deep4Net, and 22 hours 19 minutes for FBCSPNet. Inference time is also considerably shorter, with our method requiring only 55 seconds, whereas EEGNet, FBCSPNet, and Deep4Net take 613 seconds, 396 seconds, and 455 seconds, respectively. These results demonstrate that our model offers substantial reductions in computational cost, making it more efficient for real-time applications while maintaining strong performance.

Table 10: Comparison of Training and Inference Times

| Method | Training Time | Inference Time |
|---|---|---|
| EEGNet | 5 hours 33 minutes | 613 seconds |
| FBCSPNet | 22 hours 19 minutes | 396 seconds |
| Deep4Net | 9 hours 1 minute | 455 seconds |
| Ours | 2 hours 14 minutes | 55 seconds |

## J  RELIABILITY ANALYSIS OF THE POLARITY-SELECTION PROCEDURE

### J.1  POLARITY OPTIMIZATION ALGORITHM

---

**Algorithm 1:** Polarity Optimization Procedure

**Input:** Initial polarity vector $\beta$, validation sets $\mathcal{D}_1, \mathcal{D}_2$, initial loss $L_{\text{orig}}$
**Output:** Updated polarity vector $\beta$
**for** $i = 1$ **to** $N$ **do**

> Randomly sample a subset $S \subseteq \{1, \ldots, N\}$ with $|S| \in \{1, \ldots, 5\}$;
> Construct proposal $\beta'$ by flipping signs of $\beta$ on indices in $S$;
> Compute validation losses:
>
> $$L_1 = \frac{1}{|\mathcal{D}_1|} \sum_{x \in \mathcal{D}_1} \text{ValLoss}(x), \quad L_2 = \frac{1}{|\mathcal{D}_2|} \sum_{x \in \mathcal{D}_2} \text{ValLoss}(x)$$
>
> Let $L = L_1 - L_2$;
> **if** $L \leq L_{\text{orig}}$ **then**
>> Accept proposal: $\beta \leftarrow \beta'$, $L_{\text{orig}} \leftarrow L$;
>
> **else**
>> Reject proposal;

**return** $\beta$;

---

During training, the polarity-optimization routine in Algorithm 1 is invoked once every two epochs, using only the validation dataset. To avoid unnecessary computation, we employ an early-termination rule: if five consecutive invocations (corresponding to ten training epochs) result in no accepted polarity update, the optimization is considered converged. After this point, the polarity-selection module is no longer called for the remainder of training.

Each call to Algorithm 1 iterates over all $N$ nodes and evaluates one validation forward pass per proposal. In practice, the total runtime of one optimization call is comparable to that of a single training epoch. This aligns with our empirical measurements across all datasets and model variants.

## J.2 STABILITY WITH RESPECT TO INITIALIZATION

We further examine the sensitivity of the procedure to initialization. Across different train/validation splits, the empirical covariance matrices computed from training data remain highly consistent with those computed from the full dataset. After normalization, the average Frobenius distance between the two covariance matrices is $1.07 \times 10^{-4}$, which is negligibly small. Since only the signs of covariance entries are used to initialize node polarities, such minor variations do not affect the initialization. In particular, for LOSO settings where the training set constitutes approximately $99\%$ of the data, the covariance estimated from the training subset is effectively identical to that of the entire dataset. Consequently, polarity initialization remains stable, and we observe no degradation in convergence behavior or final performance.

