# OpenReview forum: "Lightweight Transformer for EEG Classification via Balanced Signed Graph Algorithm Unrolling"
_ICLR.cc/2026/Conference — ICLR 2026 Poster_

### Official Review · Reviewer_NKBD · 2025-10-31

**Soundness:** 2
**Presentation:** 3
**Contribution:** 2
**Rating:** 6
**Confidence:** 4

**Summary:**

This paper proposed an EEG classification method based on unrolling a sequence of graph learning and low-pass graph filtering operations. At each step, the authors first learn a signed graph Laplacian, then cut off the high-frequency components for denoising. The authors interpret the final output as the posterior mean of the clean signal, and derive a classifier based on its distance to the noisy input. Experiments show that their method outperforms existing graph and non-graph EEG classification models in normal and LOSO settings.

**Strengths:**

1. The paper is well-written and easy to follow. More details are provided in the appendix.
2. The authors interpret the graph learning and denoising as a form of transformers. The parameter space of the proposed model is substantially smaller than many existing methods, while the performance is stronger.

**Weaknesses:**

1. The graph construction is heuristic. Some design choices are not well-justified, e.g. the use of the Manhattan distance.
2. The evaluation on the real dataset is on the weaker side. The authors only evaluate on two datasets, with one relegated to the appendix. More informative metrics such as PR-AUC should also be considered.
3. The paper would benefit from some in-depth discussion about why the proposed model outperforms the competing ones.

**Questions:**

1. What is the justification for using the signed Laplacian? The total positivity of the combinatorial graph Laplacian is shown to be a useful bias for many problems. If both positive and negative correlations are allowed, one can just use eigenvectors of the empirical covariance matrix for denoising, essentially PCA. This is somewhat similar to the authors' choice, given their way of selecting the signs in the graph, but much simpler. Is there any ablation study on this?
2. How did the authors select the cutoff value $\omega$ for denoising?

---

> ### Author Response · Authors · 2025-11-21
> **Rebuttal by Authors Part I**
>
> We respond to the reviewer's raised weaknesses and questions one-by-one below.
>
> **Response to W1:** As shown in eq. (7), edge weights $w_{i,j}$ of our balanced signed graph are exponential functions of negative pairwise feature distances $d_{i,j}$—larger inter-node distance implies a smaller edge weight—which is common in the graph signal processing (GSP) [1]  and machine learning literature [2, 3]. We only extend its usage to balanced signed graphs as well. The use of the learned metric matrix $\mathbf{M}$ to define Mahalanobis distance is called metric learning in the machine learning literature [4], and recently adopted in the GSP community as well [5].
>
> **Response to W2:** Thank you for the suggestion regarding evaluation on real datasets and additional metrics. The Turkish Epilepsy EEG Dataset used in our main experiments is one of the most comprehensive and widely adopted benchmarks in the epilepsy-detection literature, and prior studies in EEG classification typically evaluate on only one or two datasets [6-8], so our setup follows established practice. Our method compares two denoisers through the difference in reconstruction errors, which is not a probabilistic score; thus applying a threshold to compute PR-AUC is not mathematically appropriate. Consistent with prior work, we therefore report Accuracy, Precision, Recall, Specificity, and F1-score.
>
> **Response to W3:** We thank the reviewer for this kind suggestion. We will add a paragraph discussing the comparative results in the revised manuscript.
>
>
>
> **References:**
>
> [1] A. Ortega, P. Frossard, J. Kovačević, J. M. F. Moura, and P. Vandergheynst, “Graph signal processing: Overview, challenges, and applications,” _Proceedings of the IEEE_, vol. 106, no. 5, pp. 808–828, May 2018.
>
> [2] M. Belkin and P. Niyogi, “Laplacian eigenmaps for dimensionality reduction and data representation,” _Neural Computation_, vol. 15, no. 6, pp. 1373–1396, Jun. 2003.
>
> [3] J. Shi and J. Malik, “Normalized cuts and image segmentation,” _IEEE Transactions on Pattern Analysis and Machine Intelligence_, vol. 22, no. 8, pp. 888–905, Aug. 2000.
>
> [4] K. Q. Weinberger and L. K. Saul, “Distance metric learning for large margin nearest neighbor classification,” _Journal of Machine Learning Research_, vol. 10, pp. 207–244, Feb. 2009.
>
> [5] C. Yang, G. Cheung, and W. Hu, “Signed Graph Metric Learning via Gershgorin Disc Perfect Alignment,” _IEEE Transactions on Pattern Analysis and Machine Intelligence_, vol. 44, no. 10, pp. 7219–7234, Oct. 2022.
>
> [6] Salim Rukhsar, Anil Kumar Tiwari, “Lightweight convolution transformer for cross-patient seizure detection in multi-channel EEG signals,” _Computer Methods and Programs in Biomedicine_, vol. 242, pp. 107856, 2023.
>
> [7] Oh Shu Lih, V. Jahmunah, Elizabeth Emma Palmer, Prabal D. Barua, Sengul Dogan, Turker Tuncer, Salvador García, Filippo Molinari, U Rajendra Acharya, “EpilepsyNet: Novel automated detection of epilepsy using transformer model with EEG signals from 121 patient population,” _Computers in Biology and Medicine_, vol 164, pp. 107312, 2023.
>
> [8] Albaqami, H.; Hassan, G.M.; Datta, A, “Automatic Detection of Abnormal EEG Signals Using WaveNet and LSTM,” _Sensors 2023_, vol. 23, no. 5960, 2023.

---

> ### Author Response · Authors · 2025-11-21
> **Rebuttals by Authors Part II**
>
> **Response to Q1:** We thank the reviewer for this comment. We clarify that we do not use the signed graph Laplacian $\mathbf{L}^s$, as defined in footnote 4, as our underlying graph. Instead, we first define the combinatorial graph Laplacian $\bar{\mathbf{L}}^B$, which may be indefinite, then compute a positive semi-definite (PSD) version $\bar{\mathcal{L}}^B = \bar{\mathbf{L}}^B + \delta \mathbf{I}$ by adding a scaled identity matrix $\delta \mathbf{I}$. See discussions at the end of Section 3.1 as well as Appendix C.
>
> The reviewer kindly suggested using eigenvectors of the empirical covariance matrix for denoising. While interesting, there are a few notable shortcomings compared to our proposal. First, our linear-time scheme, based on Lanczos filter approximation, never computes the full eigenvectors explicitly (with worst-case complexity of $\mathcal{O}(N^3)$) for low-pass filtering. Second, though the (sparse) inverse covariance matrix is often interpreted as the graph Laplacian (computed via methods such as graphical lasso [1]), because its off-diagonal terms can have positive and negative entries, it is an unbalanced signed graph in general, which does not have a well-defined notion of frequencies, as discussed in our paper. “Low-pass” filters for unbalanced signed graphs do not have good performance, as shown in Table 2. Finally, unlike the empirical covariance matrix, our signed graph’s pairwise relationships are not computed based purely on empirical correlations, but on feature distance based on feature vectors learned via a shallow CNN, resulting in a learned graph.
>
> Regarding ablation studies, Table 2 compares the performance of using our proposed balanced signed graph against a positive graph and an unbalanced signed graph.
>
> **Response to Q2:** We thank the reviewer for this question. The cutoff frequency $\omega$ for each ideal low-pass graph filter is learned per block from data after algorithm unrolling. Kindly see Appendix E.3 for a detailed discussion.
>
> **References:**
>
> [1] J. Friedman, T. Hastie, and R. Tibshirani, “Sparse inverse covariance estimation with the graphical lasso,” _Biostatistics_, vol. 9, no. 3, pp. 432–441, 2008.

---

### Official Review · Reviewer_ZtHH · 2025-10-31

**Soundness:** 3
**Presentation:** 3
**Contribution:** 3
**Rating:** 8
**Confidence:** 4

**Summary:**

The paper addresses the problem of EEG classification. The authors propose a discriminant
analysis model that predicts classes based on the smallest reconstruction error to a
class-wise autoencoder. The autoencoder represents the multivariate EEG time series
as a graph where each node represents a segment of a univariate channel.
Consecutive segments in the same channel are connected by edges, and concurrent
channel segments are connected by edges weighted by the correlation between their
segments. The autoencoder denoises by using a graph low-pass filter that uses only
the k eigen dimensions of the graph Laplacian with highest eigenvalue. The authors
address specifically the problem that some of the edge weights / correlations can
be negative, and then the graph Laplacian is no longer guaranteed to be positive
semidefinite. For this case they approximate the graph with a balanced signed
graph, flipping the signs of some edges; then a pos.semidef. Laplacian is guaranteed
again. In experiments on a large EEG dataset they show that their method outperforms
several baselines, while reaching 98.4% of the performance of a way less
parsimonious model with 1000 times more parameters.

**Strengths:**

- s1. denoising using the channel segment correlation graph is an interesting approach.
- s2. finding a good balanced signed graph approximation is a novel idea in this context.
- s3. very good results with a model with very few parameters.
- s4. the method is well explained.

**Weaknesses:**

- w1. several strong baseline papers for EEG classification have been missed.
- w2. an ablation study for low pass filtering is missing.
- w3. experiments are hard to reproduce as no source code is provided and the method
  is somewhat complex.

more details:

w1. several strong baseline papers for EEG classification have been missed.
- for example, MAtt [Pan et. al., NeurIPS 2022] is often compared to recently.
- also the EEG foundation models such as EEG2Rep [Baeza-Yates et al., KDD 2024]
  might be interesting models to compare to. They also report on TUAB
  (like your appendix G), but way lower scores. Is the experimental protocol
  the same?

w2. an ablation study for low pass filtering is missing.
- would the model deteriorate if you do not low-pass filter at all?
- then your model is basically a graph attention neural network.
- what is not clear to me: to softmax-like normalization of the
  attention weights in eq. 8 should denoise already by pushing
  the smaller weights close to zero. Low-pass filtering now sets them
  to exactly zero. Why does this help?
- you argue that your model does not use dense and large key,
  value and query matrices K, V and Q, but the Mahalanobis kernel M
  in eq. 6 you could interprete as say the key weights matrix Q.
  Then you just choose the identity for K and V.

w3. experiments are hard to reproduce as no source code is provided and the method
is somewhat complex.
- re-implementing all the different steps of your method will put a high burden on
  researchers trying to work with your paper.

smaller points:
- p1. the model's performance crucially depends on the contrastive loss (eq. 23),
  but you do not mention this in the main paper. In ablation study F1, w/o
  the constrastive loss your model performs worse than all your baselines
  in tab. 1.


references:
- Pan, Yue-Ting; Chou, Jing-Lun; Wei, Chun-Shu (NeurIPS 2022):
  "MAtt: a manifold attention network for EEG decoding."
  Advances in Neural Information Processing Systems.

- Baeza-Yates, Ricardo; Bonchi, Francesco; Nguyen, Nam; Foumani, Navid
  Mohammadi; Salehi, Mahsa; Mackellar, Geoffrey; Ghane, Soheila; Irtza,
  Saad (KDD 2024): "EEG2Rep: Enhancing Self-supervised EEG
  Representation Through Informative Masked Inputs."  Proceedings of the
  30th ACM SIGKDD Conference on Knowledge Discovery and Data Mining, KDD
  2024, Barcelona, Spain, August 25-29, 2024.

**Questions:**

- q1. Can you compare the performance of your model with recent strong models
  such as MAtt and EEG2Rep?
- q2. As additional ablation study: how does the model perform w/o low-pass filtering?

---

> ### Author Response · Authors · 2025-11-21
> **Rebuttal by Authors**
>
> **Response to W1:** We thank the reviewer for the recommended related works. Given the substantial time required to conduct thorough experiments, we will conduct the experiments in the next two weeks and include the results in the revised paper by December 3rd.
>
> **Response to W2:**  We thank the reviewer for this comment. We respond in parts:
> Our model depends heavily on low-pass filtering to construct a signal of a desired shape; eigenvectors corresponding to small eigenvalues are signals that are consistent with the learned similarity graph, and a low-pass filtered signal is a weighted linear combination of these eigenvectors. Thus, given an appropriately learned similarity graph capturing pairwise (anti-)correlations, the desired signal must be low-pass. The only remaining question is how strongly a low-pass filter should be applied relative to observed data; the strength of the graph filters is determined by the cutoff frequency $\omega$, which is tuned in a data-driven manner per neural layer duration back-propagation.
>
> The reviewer is correct in pointing out that softmax serves the purpose of normalization, similar in principle as our symmetric normalization of graph edge weights in eq. (8). Given normalized edge weights, a similarity graph capturing pairwise (anti-)correlations is well defined, then a signal of desired shape can be constructed via an appropriately parametrized low-pass filter.
>
> In short, ***the combination of feature learning (via shallow CNN) and Mahalanobis distance learning (defined by metric matrix $\mathbf{M}$) can be viewed as a more parameter-efficient substitute of the query and key embedding matrices $\mathbf{Q}$ and $\mathbf{K}$ in conventional self-attention***. The affinity score $e_{i,j} = \mathbf{x}_j^\top \mathbf{Q} \mathbf{K}^\top \mathbf{x}_i$ in eq.(14) is a transformed dot product computed using large and dense matrices $\mathbf{Q}$ and $\mathbf{K}$. Given that real, symmetric and PSD metric matrix $\mathbf{M}$ can be eigen-decomposed (via the Spectral Theorem) into $\mathbf{M} = \mathbf{V} \text{diag}(\{\lambda_k\}) \mathbf{V}^\top$ for $\lambda_k \geq 0, \forall k$, we can rewrite Mahalanobis distance as
>
> $$
> \begin{equation}
> d_{i,j} = (\mathbf{f}_j - \mathbf{f}_i)^\top \mathbf{H} \mathbf{H}^\top (\mathbf{f}_j - \mathbf{f}_i),
> \end{equation}
> $$
>
> where $\mathbf{H} = \mathbf{V} \text{diag}(\{\sqrt{\lambda_k}\})$. We see now that Mahalanobis distance $d_{i,j}$ closely mimics affinity score $e_{i,j}$, with the important difference that $d_{i,j}$ is computed in the learned feature space of small dimension $K$. Thus, assuming that low-dimensional feature vectors $\mathbf{f}_i$ can be computed efficiently via a shallow CNN, our Mahalanobis distance computation is much more parameter-efficient than conventional self-attention.
>
> **Response to W3**: We thank the reviewer for this comment, which we respond in parts:
> 1. We will make our code available upon acceptance of the paper.
> 2. Indeed, the performance of our method depends on the use of the contrastive loss function. We will emphasize its importance in the revised paper.
> 3. We will include suggested references in the revised paper for juxtaposition.
>
> **Response to Q1:** Please see response to W1.
>
> **Response to Q2:**  We thank the reviewer for the thoughtful suggestion. A low-pass filtered signal simply means a signal consistent with the similarity graph structure–a linear combination of “smooth” eigenvectors corresponding to small eigenvalues of the shifted graph Laplacian matrix $\bar{\mathcal{L}}^B$. A “smooth” eigenvector $\mathbf{v}$ of Laplacian $\bar{\mathcal{L}}^B$ for a balanced signed graph here means that most connected pairs $(i,j)$ satisfy $\text{sign}(v_i v_j) = \text{sign}(w_{i,j})$, i.e., in most cases, edge signs correctly describe the connected pairs’ correlations / anti-correlations in the signal [1]. Given that the balanced signed graph is designed to learn intrinsic pairwise (anti-)correlations, it only makes sense to employ low-pass filtering (as opposed to high-pass filtering) to compute signals consistent with the learned similarity graph. See Table 2 where low-pass filters of different graph constructions result in different performances.
> The only remaining question is how much low-pass filtering should be applied in each block. In our implementation, we learn the optimal cutoff frequency $\omega$ from data via back-propagation; see Section 4.1 for details.
>
> **References:**
>
> [1] C. Dinesh, G. Cheung, S. Bagheri and I. V. Bajić, "Efficient Signed Graph Sampling via Balancing & Gershgorin Disc Perfect Alignment," in _IEEE Transactions on Pattern Analysis and Machine Intelligence_, vol. 47, no. 4, pp. 2330-2348, April 2025

---

> > ### Comment · Reviewer_ZtHH · 2025-11-27
> >
> > Thanks to the authors for their answers to my questions and their willingness
> > to include the results in the final version of the paper. It confirms my score.

---

### Official Review · Reviewer_zTYv · 2025-11-04

**Soundness:** 2
**Presentation:** 3
**Contribution:** 2
**Rating:** 4
**Confidence:** 2

**Summary:**

The paper proposes a lightweight, interpretable “transformer-like” network for EEG seizure classification by unrolling a balanced signed-graph denoising algorithm. Key ideas: (i) learn a balanced signed graph of EEG channels, then map it via a similarity transform to an equivalent positive graph with well-defined frequencies; (ii) implement an ideal low-pass graph filter (cutoff learned from data) and interleave it with a balanced-graph learning (BGL) module; (iii) train two class-conditioned denoisers and classify by comparing reconstruction errors. On the Turkish Epilepsy EEG dataset the method attains 97.57% accuracy with only ~14.8k parameters, and shows strong leave-one-subject-out (LOSO) generalization.

**Strengths:**

S1. Principled modeling. The use of balanced signed graphs gives a rigorous frequency notion via a Laplacian similarity transform to a positive graph, enabling classical spectral filtering while still modeling anti-correlations that are common in EEG. This is technically neat and well-motivated.

S2. Clear algorithm-unrolling design with interpretability. The network cleanly alternates LP filtering and BGL. Figure 2 makes the pipeline easy to follow。

S3. Good parameter efficiency and competitive accuracy.

**Weaknesses:**

> W1. “Attention equivalence” is overstated.

The normalized weights use
    $$\bar w_{ij}=\beta_i\beta_j\frac{e^{-d_{ij}}}{\sqrt{\sum_\ell e^{-d_{i\ell}}\sum_k e^{-d_{kj}}}},$$
    which (a) can be negative via $\beta_i\beta_j$, (b) are *symmetric* rather than row-stochastic, and (c) need not satisfy $\sum_j \bar w_{ij}=1$. This differs materially from softmax attention
    $$a_{ij}=\frac{\exp(e_{ij})}{\sum_k \exp(e_{ik})}\ge 0,$$
    so calling $\bar w_{ij}$ “essentially attention weights” risks confusion.

> W2. Possible leakage in polarity initialization.

Initializing polarities via an empirical covariance $\bar C$ computed from “collected EEG data” is ambiguous: if $\bar C$ uses the *entire* dataset, the train/test split is compromised. Clarify that initialization uses train-only statistics.

> W3. Temporal modeling is inconsistent across sections.

One place “assumes a single chunk” and works on $G_B$, elsewhere the implementation uses a product graph with 6 windows (nodes $N=210=6\times35$). It is unclear whether the experiments use the single-chunk simplification or the 6-slice product graph with temporal edges, and how this choice affects results.

> W4. Feature metric is under-specified.

Distances use $d_{ij}=(f_i-f_j)^\top M (f_i-f_j),\quad M\succeq0,$  but Appendix text suggests $M=Q_iQ_i^\top$ selected from a candidate set. Is $M$ global, per-node, or per-block? How is the candidate set regularized to avoid degenerate metrics? What prevents collapse(e.g., rank-1 $M$) beyond the implicit CNN bottleneck?

**Questions:**

**I found that the main text exceeds the 9-page limit, which seems to violate the strict page requirements for submissions.**

For rebuttal, I hope the authors to answer the following questions:

Q1: Please clarify whether the cutoff is implemented via explicit spectral truncation or via a smooth transfer function on eigenvalues (e.g., sigmoid). If both appear in the paper, which one produced the reported numbers, and why was it preferred?

Q2: You ensure PSD by setting $L_B=\bar L_B+\delta I$. What is the empirical distribution of $\delta$ across subjects/blocks? How does varying $\delta$ shift the effective spectrum and the LP response (e.g., energy scaling)?

Q3: How imbalanced are the learned signed graphs before enforcing balance (e.g., frustration index, fraction of odd negative cycles)? A short diagnostic could justify that hard balance enforcement does not wash out meaningful anti-correlations.

Q4: Please provide pseudocode: update order, stopping criterion, and complexity per block for $\beta_i\in\{\pm1\}$ updates. Do multiple random initializations converge to the same polarity assignment, or is there sensitivity?

Q5: You mention covariance-based initialization. Can you confirm that statistics for initialization are computed only on training folds in LOSO/default splits? If not, could you re-run with train-only stats or provide a control showing negligible impact?

Q6: Your normalized weights are symmetric and can be negative via polarities, unlike standard softmax attention. Could you either (a) provide a formal mapping/conditions when your weights behave like attention, or (b) present an ablation replacing your weight construction with softmax attention to show comparable behavior?

Q7: For baseline speed/latency numbers: what hardware, precision , batch sizes, and dataloader settings were used? If possible, include on-device CPU or low-power GPU latency to substantiate the “lightweight” claim.

---

> ### Author Response · Authors · 2025-11-21
> **Rebuttal by Authors Part I**
>
> We respond to the reviewer's raised weaknesses and questions one-by-one below.
>
> **Response to W1:** We thank the reviewer for this comment on the basic question of “what constitutes attention”. We humbly point out that there exist a larger number of implementations of self-attention in the literature beyond the original scaled dot-product definition in the seminal NeurIPS’17 paper “Attention Is All You Need”. Specifically, instead of the query and key matrices, $\mathbf{Q}$ and $\mathbf{K}$, there exist other computation methods for affinity score $e_{i,j}$, including sparse attention [1], low-rank linear attention [2], kernel-based linear attention [3], memory-augmented attention [4]. Further, instead of the conventional softmax, there exists other normalization methods to $a_{i,j}$, including Entmax [5], Sparsemax [6] and Sigmoid-based attention [7]. ***All of the above are generally considered self-attention mechanisms, because they all compute context-dependent weights to combine token representations, where the weights depend on pairwise interactions within the same sequence.*** In that sense, our graph edge weights computation is no different, and thus can be rightfully called “self-attention weights” also.
>
> **Response to W2:** We thank the reviewer for this comment on node polarity initialization. The empirical covariance matrix $\mathbf{C}$ computed from the entire dataset versus one computed from the training set are very similar: the average Frobenius distance is 0.000107. Since only the signs are used for polarity initialization, this difference does not affect the initialization at all. In LOSO, the training set covers 99% of the data, so train-only covariance is almost the same as full-data covariance, yielding identical initialization.
>
> **Response to W3:** The polarity initialization is computed from a single complete EEG segment (length 6000) and then replicated six times to initialize the nodes of the 6-chunk product graph, reflecting the expectation that a node and its temporal copies within a short window share the same polarity. Each EEG segment is divided into six temporal chunks (length 1000 each), which are assembled into the product graph used as input for every iteration.
>
> **Response to W4:** We first note that any symmetric positive semi-definite (PSD) matrix $\mathbf{M}$ is eigen-decomposible to $\mathbf{M} = \mathbf{V} \text{diag}(\{\lambda_k\}) \mathbf{V}^\top$ via the Spectral Theorem, where $\mathbf{V}$ contains orthonormal eigenvectors and $\lambda_k \geq 0, \forall k$ are non-negative eigenvalues. Thus, any symmetric PSD $\mathbf{M}$ can be written as $\mathbf{M} = \mathbf{Q} \mathbf{Q}^\top$, where $\mathbf{Q} = \mathbf{V} \text{diag}(\{\sqrt{\lambda_k}\})$, and learning $\mathbf{Q}$ is equivalent to learning a symmetric PSD $\mathbf{M}$.
> One symmetric PSD metric matrix $\mathbf{M}$ (or equivalently $\mathbf{Q}$ is learned each time a graph learning module is executed, i.e., the BGL blocks shown in Fig. 2. During learning, we initialize the matrix $\mathbf{Q}$ as a randomly generated real matrix, and learn individual entries one-by-one via stochastic gradient descent (SGD). We have revised the paper to explicitly state this initialization strategy. A rank collapse will not be a possible outcome, because it would lead to feature distance $d_{i,j} = 0$ and edge weight $w_{i,j} = 1$ for same-polarity node pair $(i,j)$ ($\beta_i = \beta_j$). This means an unweighted signed graph, which in general has worse performance than a more informative weighted signed graph in signal denoising.
>
> **References:**
>
> [1] Zaheer, M., Guruganesh, G., Dubey, A., et al., “Big Bird: Transformers for Longer Sequences,” _Advances in Neural Information Processing Systems_, vol. 33, pp. 17283–17297, 2020.
>
> [2] Wang, S., Li, B. Z., Khabsa, M., Fang, H., and Ma, H., “Linformer: Self-Attention with Linear Complexity,” _arXiv preprint_, arXiv:2006.04768, 2020.
>
> [3] Choromanski, K., Likhosherstov, V., Dohan, D., et al., “Rethinking Attention with Performers,” _International Conference on Learning Representations (ICLR)_, 2021.
>
> [4] Dai, Z., Yang, Z., Yang, Y., Carbonell, J., Le, Q., and Salakhutdinov, R., “Transformer-XL: Attentive Language Models Beyond a Fixed-Length Context,” _Proc. 57th Annual Meeting of the Association for Computational Linguistics_, pp. 2978–2988, 2019.
>
> [5] Martins, A. F. T., Farinhas, A., Treviso, M., Niculae, V., Aguiar, P. M. Q., and Figueiredo, M. A. T., “Sparse and Continuous Attention Mechanisms,” _Advances in Neural Information Processing Systems (NeurIPS)_, 2020.
>
> [6] Martins, A. F. T., and Astudillo, R. F., “From Softmax to Sparsemax: A Sparse Model of Attention and Multi-Label Classification,” _Proc. 33rd International Conference on Machine Learning (ICML)_, pp. 1614–1623, 2016.
>
> [7] Ramapuram, J., Danieli, F., Dhekane, E., et al., “Theory, Analysis, and Best Practices for Sigmoid Self-Attention,” _International Conference on Learning Representations (ICLR)_, 2025.

---

> ### Author Response · Authors · 2025-11-21
> **Rebuttal by Authors Part II**
>
> **Response to Q1:** We thank the reviewer for the comment on the implementation of the learned cutoff frequency $\omega$. In Appendix E.3, we describe that the cutoff is implemented via a differentiable sigmoid function rather than an explicit spectral truncation. Specifically, the frequency response function is defined as $g(\lambda_i) = \sigma\big( \alpha (\omega - \lambda_i) \big)$, where $\sigma$ is the sigmoid function and $\alpha$ controls the smoothness. This approach allows end-to-end differentiation with respect to $\omega$. In the revised version, we will clarify in the main text that the reported numbers are obtained using this sigmoid-based cutoff.
>
> **Response to Q2:** We thank the reviewer’s comment on the spectrum of positive graph Laplacian $\mathcal{L}^+ = \mathbf{T} \mathcal{L}^B \mathbf{T}^{-1}$, where shifted graph Laplacian matrix is $\mathcal{L}^B = \bar{\mathbf{L}}^B + \delta \mathbf{I}$. We first note that adding a scaled identity matrix $\delta \mathbf{I}$ to $\bar{\mathbf{L}}^B$ increases each eigenvalue $\lambda_k$ of $\bar{\mathbf{L}}^B$ by $\delta$, while the eigenvectors $\{\mathbf{v}_k\}$ of $\bar{\mathbf{L}}^B$ remains unchanged. $\mathcal{L}^+$ and $\mathcal{L}^B$ share the same eigenvalues, since they are similarity transforms of each other. Eigenvectors of $\mathcal{L}^+$ are simple sign flips of eigenvectors of $\mathcal{L}^B$ based on node polarities.
>
> An ideal low-pass filter for spectrum of $\mathcal{L}^+$ with cutoff frequency $\omega$ means projecting a signal onto the subspace $\mathcal{S}_\omega$ spanned by the first $\omega$ eigenvectors of $\mathcal{L}^+$ (see Section 3.2). Since the eigenvectors are the same for $\bar{\mathbf{L}}^B$ and $\mathcal{L}^B$ regardless of $\delta$, eigenvectors of $\mathcal{L}^+$ are also unaffected by $\delta$. Thus, the low-pass filter for the spectrum of $\mathcal{L}^+$ remains unchanged regardless of $\delta$, though the spectrum (set of eigenvalues) itself has shifted up by $\delta$.
>
> Shifting $\bar{\mathbf{L}}^B$ to PSD $\mathcal{L}^B$ by adding $\delta \mathbf{I}$ makes the augmented non-negative eigenvalues $\{\lambda_k\}$ more easily interpretable as graph frequencies. Further, practically, the Lanczos filter approximation–a Krylov method–is more stable when the Laplacian matrix is PSD, since the Ritz values (eigenvalues of $\mathbf{H}_M$ in eq. (24)) converge more predictably [1].
>
> **Response to Q3:** We thank the reviewer for this question on graph balance. While not a rigorous mathematical proof, there is empirical evidence that many datasets in the real world with strong anti-correlations do tend towards graph balance, including social, biological, economic, and collaborative networks [2-5]. For example, two people A and B having a mutual friend C are likely to be friends themselves, resulting in a balanced triad (A, B, C). For our specific EEG data, we divided the training signals along the time dimension into six chunks and constructed a graph structure identical to the Positive Product Graph shown in Figure 1. Edge weights were set according to the corresponding entries in the average covariance matrix computed from node signals. By setting a threshold of -0.1, edges with weights below this value were considered as indicating strong anti-correlation between the corresponding nodes, and were thus treated as negative edges. Analyzing the resulting graph, we observed 6 odd negative basic cycles out of 1481 basic cycles checked, corresponding to a fraction of 0.004. Furthermore, applying a greedy node-flip heuristic [6] to estimate the frustration index yielded a value of 16 out of 1322 edges. These results indicate that the original signals already exhibit a largely balanced structure. Therefore, enforcing balance has minimal impact on the meaningful anti-correlation patterns present in the data.
>
> **References:**
>
> [1] C. Musco, C. Musco, and A. Sidford, "Stability of the Lanczos Method for Matrix Function Approximation," in Proceedings of the 29th _Annual ACM-SIAM Symposium on Discrete Algorithms (SODA)_, 2018, pp. 2487–2506.
>
> [2] T. Antal, P. L. Krapivsky, and S. Redner, “Dynamics of social balance on networks,” _Phys. Rev. E_, vol. 72, no. 3, p. 036121, Sep. 2005.
>
> [3] J. Hu and W. X. Zheng, “Bipartite consensus for multi-agent systems on directed signed networks,” in _52nd IEEE Conference on Decision and Control_, Dec. 2013, pp. 3451–3456.
>
> [4] C. M. Rawlings and N. E. Friedkin, “The Structural Balance Theory of Sentiment Networks: Elaboration and Test,” _American Journal of Sociology_, vol. 123, no. 2, pp. 510–548, Sep. 2017.
>
> [5] A. Gallo, D. Garlaschelli, R. Lambiotte, F. Saracco, and T. Squartini, “Testing structural balance theories in heterogeneous signed networks,” _Commun Phys_, vol. 7, no. 1, pp. 1–13, May 2024.
>
> [6] Angela Fontan, Marco Ratta, Claudio Altafini, “From populations to networks: Relating diversity indices and frustration in signed graphs,” _PNAS Nexus_, vol. 3, no. 2, pp. 46, February 2024.

---

> ### Author Response · Authors · 2025-11-21
> **Rebuttal by Authors Part III**
>
> **Response to Q4:** We thank the reviewer for requesting clarification on the update procedure. The polarity optimization follows the pseudocode presented below, which summarizes the update order and acceptance rule:
> ```pseudo
> Procedure OPTIMIZE-BETA(beta, D1, D2, initial_loss):
>     For  node i = 1 … N do:
>         Randomly select a small subset S of polarity indices with 1-5 nodes
>         Propose a polarity flip on S:
>             beta' ← beta with signs of indices in S flipped
>         Evaluate validation loss on both classes:
>             L1 ← average ValidationLoss(x) over x ∈ D1
>             L2 ← average ValidationLoss(x) over x ∈ D2
>         L ← sum(L1 − L2)
>         Accept polarity flip if L ≤ initial_loss else Reject
> ```
> * **Update order and stopping criterion**. During training, we run the polarity optimization once every two epochs using the validation dataset. The optimization terminates early if five consecutive proposals are rejected, i.e., no accepted polarity flip occurs across five update attempts.
> * **Complexity per block**. Each optimization step iterates over N nodes and requires one forward pass on the validation set for each polarity proposal. In practice, the runtime is comparable to one additional training epoch, as confirmed by our empirical measurements.
> * **Sensitivity to initialization**. We observe little sensitivity to initialization. Across different train/validation splits, the empirical covariance matrices remain highly consistent. Since only the **signs** are used for polarity initialization, this difference does not affect the initialization at all which lead to consequently stable convergence of the optimization procedure. Because this data-driven prior provides reliable initialization, using a random initialization would be inappropriate and would introduce unnecessary variability.
>
> **Response to Q5:** We thank the reviewer’s comment regarding node polarity initialization based on the empirical covariance matrix. To investigate this, we compared the covariance matrix computed from the training dataset with that computed from the entire dataset. We found that the matrices are almost identical: after normalization, the average Frobenius distance between the two covariance matrices is 0.000107, which is negligibly small. Furthermore, since we only use the signs of the covariance values to initialize node polarities, this minor difference does not affect the initialization. In particular, for LOSO tests, the training set accounts for 99% of the data (with only a single patient held out for testing), so the covariance computed on the training data is almost identical to that of the full dataset, and the resulting polarity initialization is essentially unchanged.
>
> **Response to Q6:** We thank the reviewer’s comment on the question of what constitutes attention. Please see our response to W1.
>
> **Response to Q7:** We thank the reviewer’s comment on the meaning of “lightweight”. To our knowledge, “lightweight models” in the deep learning literature commonly refers to models with few parameters, with desirable implications such as smaller memory footprints, lower computation cost (fewer FLOPS) [1]. Lightweight models are important for hardware-constrained end-devices with limited memory and compute power [2]. We have already demonstrated our model’s parameter efficiency definitively in Table 1. In Appendix I, we also show our model’s advantage in training time and inference time compared to competing schemes.
>
> **References:**
>
> [1] Hou-I Liu, Marco Galindo, Hongxia Xie, Lai-Kuan Wong, Hong-Han Shuai, Yung-Hui Li, and Wen-Huang Cheng. "Lightweight Deep Learning for Resource-Constrained Environments: A Survey", _ACM Comput_, Surv. 56, 10, Article 267, October 2024.
>
> [2] A. Musa, H. A. Kakudi, M. Hassan, M. Hamada, U. Umar, and M. L. Salisu, “Lightweight Deep Learning Models for Edge Devices—A Survey", _Int. J. Comput. Inf. Syst. Ind. Manag. Appl._, vol. 17, pp. 189–206, Jan. 2025.

---

### Official Review · Reviewer_hLos · 2025-11-06

**Soundness:** 3
**Presentation:** 3
**Contribution:** 2
**Rating:** 4
**Confidence:** 3

**Summary:**

The paper tackles binary classification of EEG signals. For each of the two classes, a graph-based signal denoiser is learnt from data. Denoising is done by applying a low-pass filter on the Laplacian of a learnt graph. For inference, the data point is denoised by both denoisers and the datapoint is classified as belonging to the class of the denoiser that best reconstructed the input signal. The model is then validated on real-world data where the authors highlight that it is competitive with SOTA, while employing fewer parameters.

**Strengths:**

- The paper investigates an interesting, real-world problem. Specifically in the medical domain, an interpretable model is often preferable to a stronger black box model.

- The decisions made in the design of the approach are described nicely and backed by theory and literature references

**Weaknesses:**

- Notation is abused quite heavily throughout the paper. E.g.:
  - $\Psi_r$ are the parameters of the $r$-th block. At the same time $\Psi_0(y)$ is the output of the denoiser for class $0$. Additionally, the parameters are $\Phi_r$ in Figure 2.
  - $x \in \mathbb{R}^N$ is the input signal. At the same time $x_i \in \mathbb{R}^E$ is the input embedding.
  - $c_i$ is the center of the Gersgorin disk and $c$ is also the random variable referring to the class.

- Table 1 (the main results table) is quite confusing.
  - Why are the large models in their own category?
  - The time needed for training is a much better metric for praticioners, even the 11mil model takes up less that 100MB of space, however, it will take considerably longer to train.
  - You reference Li et. al in your main text, but surely ment Bhandage et. at.

- The empirical evaluation is limited to one dataset (in two dífferent settings). The dataset that was used was essentially solved by Bhandage et. at. While the proposed method is competitive with the other baselines, there remains a gap to SOTA. In the LOSO problem setting, Bhandage et. at. are left out of the evaluation. The second experiment is quite weak in itself. Essentially, the hypothesis that "being able to encode negative correlation is advantageous" is empirically validated. This does little to strengthen the approach. No error margins are reported.

- The idea of a interpretation of the model is only hinted at in the beginning and then left underexplored. In the life sciences, an explainable model often beats a stronger black-box model because it offers insight into the mechanisms at play. The paper has a strong entry point into such an analysis, as the approach actually constructs a model of the signal distribution for each class. That is, inspecting these models could lead to an analysis of what part of the signal is responsible for the distinction.

**Questions:**

- Why are the large models in their own category in Table 1?
- How does training time of your model compare to the large models (Li, Bhandage)?
- Do you have empirical evidence on other datasets?
- Can you comment on the error margins of the approaches?
- Can you extract useful information from your models (Interpretability, Explainability)?

---

> ### Author Response · Authors · 2025-11-21
> **Rebuttal by Authors Part I**
>
> We respond to the reviewer's raised weaknesses and questions one-by-one below.
>
> **Response to W1:** We thank the reviewer for their careful reading and for pointing out the issues in our notation.
> Following the reviewer’s comments, we conducted a thorough pass over the entire manuscript and revised all ambiguous or overloaded symbols. All corrections are reflected in the revised version of the paper, where we have ensured that each symbol is used consistently and carries a unique meaning throughout the paper. Importantly, we have carefully verified that these revisions concern **presentation only** and do not affect any experimental results, implementation details, or theoretical derivations in the paper. In particular, regarding the notation issues explicitly highlighted by the reviewer:
> * $\Phi_\tau$ is set to represent the parameters of the r-th block aligned with Figure 2, and $\Psi_c(\cdot)$ is kept for the denoiser.
> * $\mathbf{e}_i \in \mathbf{R}^E$ is set to represent the input embedding aligned with the subsection entitled “Feature Distance” in Section 3.1 and $\mathbf{x}$ denotes the input signal.
> * $center_i$ denotes the center of the Gershgorin disk and $c$ is used for the random variable referring to the class.
>
> **Response to W2:** We thank the reviewer for the valuable feedback regarding the organization of the main results table.
> As our work focuses on lightweight EEG models (i.e., small parameter size), we grouped lightweight approaches into a single category to clearly emphasize that our method achieves state-of-the-art performance within this regime. Large-scale models were presented separately to provide contextual comparison, illustrating that our model—despite having orders of magnitude fewer parameters—attains accuracy comparable to substantially heavier architectures.
>
> The reviewer also raised an important point regarding training and inference time. We clarify that “lightweight” in prior studies consistently refers to models with significantly fewer parameters (e.g., [1-3]). In addition, we have already reported training and inference time in **Appendix I** for baselines with publicly available implementations, evaluated under unified settings. The method by Bhandage et al. is discussed in the description column; however, since its code has not been released, its runtime cannot currently be benchmarked. We will include a complete timing comparison once the official implementation becomes available.
>
> The citation for Bhandage et al. has also been corrected accordingly in the revised manuscript.
>
> **References:**
>
> [1] Cunhang Fan, Jinqin Wang, Wei Huang, Xiaoke Yang, Guangxiong Pei, Taihao Li, Zhao Lv, “Light-weight residual convolution-based capsule network for EEG emotion recognition,” _Advanced Engineering Informatics_, vol. 61, 2024.
>
> [2] Zhige Chen, Rui Yang, Mengjie Huang, Fumin Li, Guoping Lu, Zidong Wang, “EEGProgress: A fast and lightweight progressive convolution architecture for EEG classification,” _Computers in Biology and Medicine_, vol. 169, pp. 1-10, 2024.
>
> [3] Pengfei Hou, Xiaowei Li, Jing Zhu, Bin Hu, “A lightweight convolutional transformer neural network for EEG-based depression recognition,” _Biomedical Signal Processing and Control_, vol. 100, Part A, 2025.

---

> ### Author Response · Authors · 2025-11-21
> **Rebuttal by Authors Part II**
>
> **Response to W3:**  We thank the reviewer for these thoughtful comments. Our primary experiments are conducted on the Turkish Epilepsy EEG Dataset, one of the most comprehensive and widely used benchmarks for epilepsy detection, providing a solid and representative evaluation setting (e.g. [1-5]). In addition, as already included in our submitted manuscript, **Appendix G** reports experiments on the TUH Abnormal EEG Corpus, where we present both the detailed setup and the corresponding results to further demonstrate the robustness of our method. Regarding the concern about the second experimental design, our use of a leave-one-subject-out (LOSO) protocol is intentional, as it more faithfully reflects real clinical deployment scenarios than simple random splits and offers a stricter measure of cross-subject generalization. Finally, the reviewer’s point on error reporting is addressed through the statistical analyses already provided in **Appendix H**, where we perform a t-test for the default setting and an ANOVA test for the LOSO setting. These results indicate that the observed performance differences are statistically stable and support the reliability of our conclusions.
>
> **Response to W4** We thank the reviewer for this comment on model interpretability. “Interpretability” in our graph algorithm unrolling context is notably beneficial for the developer.  Specifically, our constructed network is “interpretable” in two major respects. First, as explained in footnote 1, “interpretability” in the deep algorithm unrolling literature typically means that each neural layer corresponds to a particular iteration of an iterative algorithm minimizing a mathematically defined objective. The practical implication is that, understanding the application-specific operations of each neural layer, a developer can more easily parametrize modules in each layer for data-driven optimization via back-propagation. For example, understanding that feature distance $d_{i,j}$ is used to compute edge weight $w_{i,j}$ in a similarity graph in eq. (7), we introduced a positive semi-definite (PSD) metric matrix $\mathbf{M}$ in layers corresponding to feature learning to compute Mahalanobis distance $d_{i,j} = (\mathbf{f}_j - \mathbf{f}_i)^\top \mathbf{M} (\mathbf{f}_j - \mathbf{f}_i)$, generalizing from Euclidean distance. Doing so means we can introduce parameters in a well-informed and controlled manner, resulting in dramatic overall parameter reduction, as we demonstrated definitively in Table 1.
>
> Second, “interpretability” in our balanced signed graph algorithm unrolling specifically means also that our implemented graph filters are low-pass with analytical filter response. This provides a useful sanity check during development; each filtered signal is expected to follow the low-pass frequency profile dictated by the graph filter. If it does not, then the Lanczos approximation filter implementation and learned cutoff frequencies should be checked. If the signal does follow the low-pass frequency profile but the denoising performance remains poor, then the constructed similarity graph is problematic, and components such as shallow CNNs used to define feature vectors $\mathbf{f}_i$’s and the learned metric matrix $\mathbf{M}$ should be checked. Summarizing, interpretability means practically that the development can be more easily tractable and debugged, and the developed network can be more efficiently parametrized.
>
> **Response to Q1:** Please see response to W2.
>
> **Response to Q2:** Please see response to W2.
>
> **Response to Q3:** Please see response to W3.
>
> **Response to Q4:** Please see response to W3.
>
> **Response to Q5:** Please see response to W4.
>
> **References:**
>
> [1] V. Bhandage, T. Pokuri, D. Desai and A. Jeyabose, "Detection of Epilepsy Disorder Using Spectrogram Images Generated From Brain EEG Signals," in _IEEE Access_, vol. 12, pp. 195054-195064, 2024.
>
> [2] Wang, J., Ge, S., Xu, H. et al. A novel binary data classification system based on the modified Gray–Scott model. _Nonlinear Dyn_, vol. 113, pp. 27659–27690, 2025.
>
> [3] Oh Shu Lih, V. Jahmunah, Elizabeth Emma Palmer, Prabal D. Barua, Sengul Dogan, Turker Tuncer, Salvador García, Filippo Molinari, U Rajendra Acharya, “EpilepsyNet: Novel automated detection of epilepsy using transformer model with EEG signals from 121 patient population,” _Computers in Biology and Medicine_, vol 164, pp. 107312, 2023.
>
> [4] R. Jain, "Seizure Detection From EEG Signals: Leveraging Poincaré Plots with Autoencoder-Classifier Model," 2025 25th _International Conference on Digital Signal Processing (DSP)_, pp. 1-5, 2025.
>
> [5] Dişli, F.; Gedikpınar, M.; Fırat, H.; Şengür, A.; Güldemir, H.; Koundal, D. "Epilepsy Diagnosis from EEG Signals Using Continuous Wavelet Transform-Based Depthwise Convolutional Neural Network Model", _Diagnostics_, vol. 15, no. 84, 2025.

---

### Author Response · Authors · 2025-12-04
**Overall Response and Key Revisions**

We thank all reviewers for their thoughtful comments. In response, we have made the following major changes in our revised paper (All modifications in the latest revision of the paper are highlighted in blue) .

In response to Reviewer hLos, we clarified that we conducted our primary experiments on the Turkish Epilepsy EEG Dataset, the most comprehensive dataset publicly available for epilepsy patients, which is consistent with the literature on this topic. Additional experiments on the TUH Abnormal EEG Corpus are discussed in Appendix G, and statistical analysis is discussed in Appendix H. We clarified also that, in the algorithm unrolling literature, “interpretability” commonly means that each neural layer can be interpreted as an iteration of an optimization algorithm minimizing a mathematically-defined objective.

In response to Reviewer zTYv, we clarified that in the machine learning literature, “sef-attention” broadly means context-dependent weights to combine token representations, where the weights depend on pairwise interactions within the same sequence. In this sense, our graph edge weights are indeed a self-attention mechanism. We also clarified our implementation of polarity initialization, learned cutoff frequency for each low-pass graph filter, and metric learning in Mahalanobis distance. We also performed analysis to show that the empirical covariance matrix is already mostly balanced (i.e., low frustration index), and our graph balancing procedure does not cancel out major anti-correlations in the collected natural statistics.

In response to Reviewer ZtHH, we compared against two additional recent EEG classification methods, MAtt and EEG2Rep, demonstrating the competitiveness of our proposal while reducing the number of parameters significantly.

In response to Reviewer NKBD, we clarified the importance of using a spectrally shifted combinatorial graph Laplacian matrix to define graph frequencies, so that a Lanczos approximation filter can be efficiently implemented.

---

### Meta-Review · Area_Chair_zZU5 · 2026-01-08

**Summary:**

The paper proposes a EEG classification method using balanced signed graph representation of EEG signals and graph-based denoising. An epilepsy EEG dataset is used for evaluation, showing efficiency of the proposed method.

**Strengths**

1. The method is principled and well-motivated.

2. The idea of finding balanced signed graph approximation is interesting.

3. The method is efficient.

**Weaknesses**

1. The evaluation is done only one dataset (and an additional one in Appendix) (hLos, NKBD). The performance is behind the SOTA method (STFT+CNN).

2. Interpretability is mentioned at the beginning but underexplored (hLos).

3. It is not clear whether initialization uses both training and test data (zTYv).

4. Strong baselines for comparison are missing (ZtHH).

**Reviewer Concerns:**

1. The rebuttal points out that Appendix G additionally contains an experiment on the TUH dataset. However, AC thinks that it would have been better to present both experiments at the same time to draw comprehensive interpretations and conclusions. In addition, the way the experiments were done on each of the two datasets doesn't seem consistent, e.g., with different baselines.

2. The rebuttal clears the meaning of interpretability.

3. The rebuttal does not explicitly confirm that initialization is done only using training data to prevent leakage, but only says that there is only little difference between initializations using all data and only training data. Thus, the reviewer's concern still remains and the reported results may not be 100% rigorous.

4. The revised paper includes the two methods mentioned by the reviewer.

**Reviewer Scores:**

Reviewers hLos might have changed the score.

Reviewer zTYv wouldn't have changed the score due to the still remaining concern about initialization.

Reviewer ZtHH would remain positive.

Reviewer NKBD replied that s/he keeps the score.

---

### Decision · Program_Chairs · 2026-01-26

Accept (Poster)